# Enabling preprint discovery, evaluation, and analysis with Europe PMC

**Mariia Levchenko** \*, **Michael Parkin, Johanna McEntyre, Melissa Harrison**

European Molecular Biology Laboratory, European Bioinformatics Institute (EMBL-EBI), Wellcome Genome Campus, Hinxton, United Kingdom

\* levchmar@ebi.ac.uk

**Data Availability Statement:** Underlying datasets and full description of the methodology documented in an R Markdown file are available from Zenodo database (DOI 10.5281/zenodo. 10711385).

## Abstract

Preprints provide an indispensable tool for rapid and open communication of early research findings. Preprints can also be revised and improved based on scientific commentary uncoupled from journal-organised peer review. The uptake of preprints in the life sciences has increased significantly in recent years, especially during the COVID-19 pandemic, when immediate access to research findings became crucial to address the global health emergency. With ongoing expansion of new preprint servers, improving discoverability of preprints is a necessary step to facilitate wider sharing of the science reported in preprints. To address the challenges of preprint visibility and reuse, Europe PMC, an open database of life science literature, began indexing preprint abstracts and metadata from several platforms in July 2018. Since then, Europe PMC has continued to increase coverage through addition of new servers, and expanded its preprint initiative to include the full text of preprints related to COVID-19 in July 2020 and then the full text of preprints supported by the Europe PMC funder consortium in April 2022. The preprint collection can be searched via the website and programmatically, with abstracts and the open access full text of COVID-19 and Europe PMC funder preprint subsets available for bulk download in a standard machine-readable JATS XML format. This enables automated information extraction for large-scale analyses of the preprint corpus, accelerating scientific research of the preprint literature itself. This publication describes steps taken to build trust, improve discoverability, and support reuse of life science preprints in Europe PMC. Here we discuss the benefits of indexing preprints alongside peer-reviewed publications, and challenges associated with this process.

## Introduction

### Preprint adoption in the life sciences

Traditionally, scientific results are communicated by publishing a peer-reviewed journal article. However, the current publishing process has a number of challenges and limitations. Increasing costs to publish open access, and to access the published literature, present a problem for both authors and readers [1]. The publishing process itself can be quite lengthy. It can

**Funding:** This work was supported by the European Molecular Biology Laboratory (EMBL). Funding for Europe PMC is provided by 35 funders of life science research under Wellcome Trust 10.35802/221523, awarded to European Molecular Biology Laboratory - European Bioinformatics Institute (EMBL-EBI). Funding for full text COVID-19 preprints in Europe PMC is supported by Wellcome 10.35802/221558 in partnership with the UK Medical Research Council (MRC) and the Swiss National Science Foundation (SNSF). The funders had no role in study design, data collection and analysis, decision to publish, or preparation of the manuscript. Grant details Europe PMC 2021-2026. Dr. Johanna McEntyre, European Bioinformatics Institute Grant ID: 221523 Grant DOI: 10.35802/221523 Full text COVID-19 preprints in Europe PMC Dr. Johanna McEntyre, European Bioinformatics Institute Grant ID: 221558 Grant DOI: 10.35802/221558.

**Competing interests:** The authors have declared that no competing interests exist.

take several months, and in some cases years, from submission to publication [2]. If the initial submission is rejected, additional time and effort is needed to re-submit to another journal. Editorial and peer review decisions can be subjective [3] and subject to gender, reputational, and diversity biases [4–7]. If mistakes, or even fraud, take place they are often hard to correct. Studies suggest that many publications are not retracted when they should be, despite an increase in publication retractions in recent years [8–10].

Preprints can provide a way to address some of these limitations. Preprints are complete scientific manuscripts without journal-organised peer review, uploaded by the authors to a public server [11]. Preprints offer several advantages, such as rapid communication and no charge to read and post. Preprints also provide an opportunity for early feedback and additional exposure prior to final publication in a journal [12, 13].

In some disciplines, such as physics, preprints have been part of research communication for decades [14]. Efforts to promote preprints in biology trace back several decades [15, 16]. Nonetheless, in the life sciences, it is still a relatively new path to share research findings. Preprints represented less than 1% of the total number of journal articles in the life sciences published in 2016 (Fig 1). Emergence of life science-specific preprint servers supported preprint adoption in the community [17–19]. Changes to publisher and funder policies have further incentivised preprinting [19]. Since 2016 several research funding organisations have opted to take preprints into consideration in funding decisions, and currently encourage or require investigators to post preprints prior to publication [19]. A 2020 study on peer review and preprint policies across a subset of major journals found that in the life and earth sciences 91% of surveyed journals would accept preprinted submissions [20], and many journals actively encourage it (https://asapbio.org/journal-policies). Several journals make manuscripts available to read before journal-organised peer review [17, 21–23]. Such publications can therefore also be considered preprints. More recently, the 'Publish, Review, Curate' (PRC) peer-review model has been pioneered by eLife and later adopted by GigaByte Journal [24, 25]. In this model every manuscript sent for review is published as a "Reviewed Preprint" and is available alongside an editorial assessment, public reviews, and a response from the authors. Other innovative publisher practices that incentivise preprinting include exclusively reviewing preprints, extended scoop protection or publication charge discounts (https://asapbio.org/journal-policies). Finally, preprint adoption by life science researchers has been heavily supported through advocacy work. Organisations such as ASAPbio have promoted the use of preprints for research dissemination.

As preprints became more established in the life sciences their growth rate increased significantly. Between 2016 and 2018, the number of life science preprints was growing 10 times faster than the number of journal articles [19, 26]. In 2023 the share of life science preprints had grown to 10.7% (Fig 1).

**Growth of preprints during the COVID-19 pandemic.** Preprints enable swift dissemination of scientific findings. Therefore, their importance becomes increasingly apparent in public health crises. Some funders and institutions, including the Bill & Melinda Gates Foundation, Médecins Sans Frontières, the US National Institute of Health, and Wellcome, have explicitly encouraged preprinting during outbreaks caused by Zika and Ebola viruses. However, preprint adoption during these epidemics remained relatively limited [27].

The COVID-19 pandemic posed a truly unprecedented challenge for scholarly communications. It required rapid sharing of research results to inform policy and clinical decision making. In 2020 stakeholders, such as the World Health Organization (WHO), Organisation for Economic Co-operation and Development (OECD), national science and technology advisors, and funding institutions, encouraged immediate open sharing of COVID-19 related publications and data [28, 29]. Preprints played an important role in early dissemination of COVID-

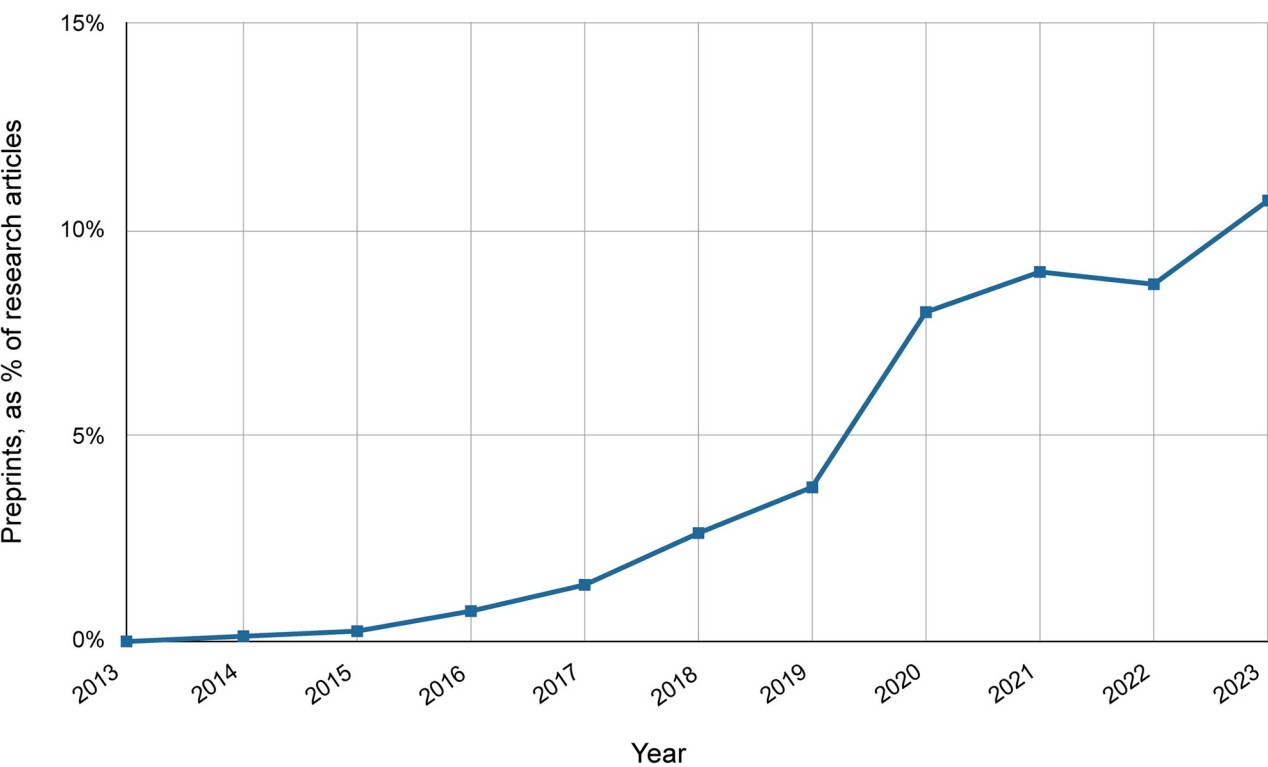

**Fig 1. Percentage of preprints in Europe PMC posted yearly between 2013–2023.** For each year the number of preprints in Europe PMC is divided by the total number of research articles, including preprints and PubMed journal articles published in the same period. Accessed 8 April 2024.

19 science. Within the first two months of the pandemic they comprised over 30% of the published COVID-19 research (Fig 2A). This was largely due to the higher speed of sharing preprints compared to journal publications despite many in the scholarly publishing community supporting this paradigm. Roughly 2–7% of COVID-19 journal articles published in 2020–2023 were preceded by a preprint (Fig 2B). Publishers' efforts included implementation of an expedited peer review and publication process, waiver of author processing charges (APCs), and open access to publications on COVID-19 [30]. In March 2020 more than 50 publishers made COVID-19 full text articles available via PubMed Central (PMC), and through PMC International, via Europe PMC (https://www.ncbi.nlm.nih.gov/labs/pmc/about/covid-19/). Several funding agencies also supported preprint adoption during the COVID-19 pandemic, encouraging or requiring research results to be shared as preprints prior to publication (https://asapbio.org/funder-policies).

### Facilitating preprint discovery with Europe PMC

Despite their significance, preprints can be harder to find and reuse than journal articles. Preprints are typically scattered across many platforms. Over 60 preprint servers have a biomedical or medical scope (https://asapbio.org/preprint-servers). Preprint coverage across different scholarly discovery engines greatly varies. One dataset reports coverage under 80% surveyed across various tools for at least 30% of preprint servers (https://docs.google.com/spreadsheets/d/1ZiCUuKNse8dwHRFAyhFsZsl6kG0Fkgaj5gttdwdVZEM/edit?gid=1016151070#gid=1016151070). Many of the discovery platforms are proprietary. Such platforms may lack

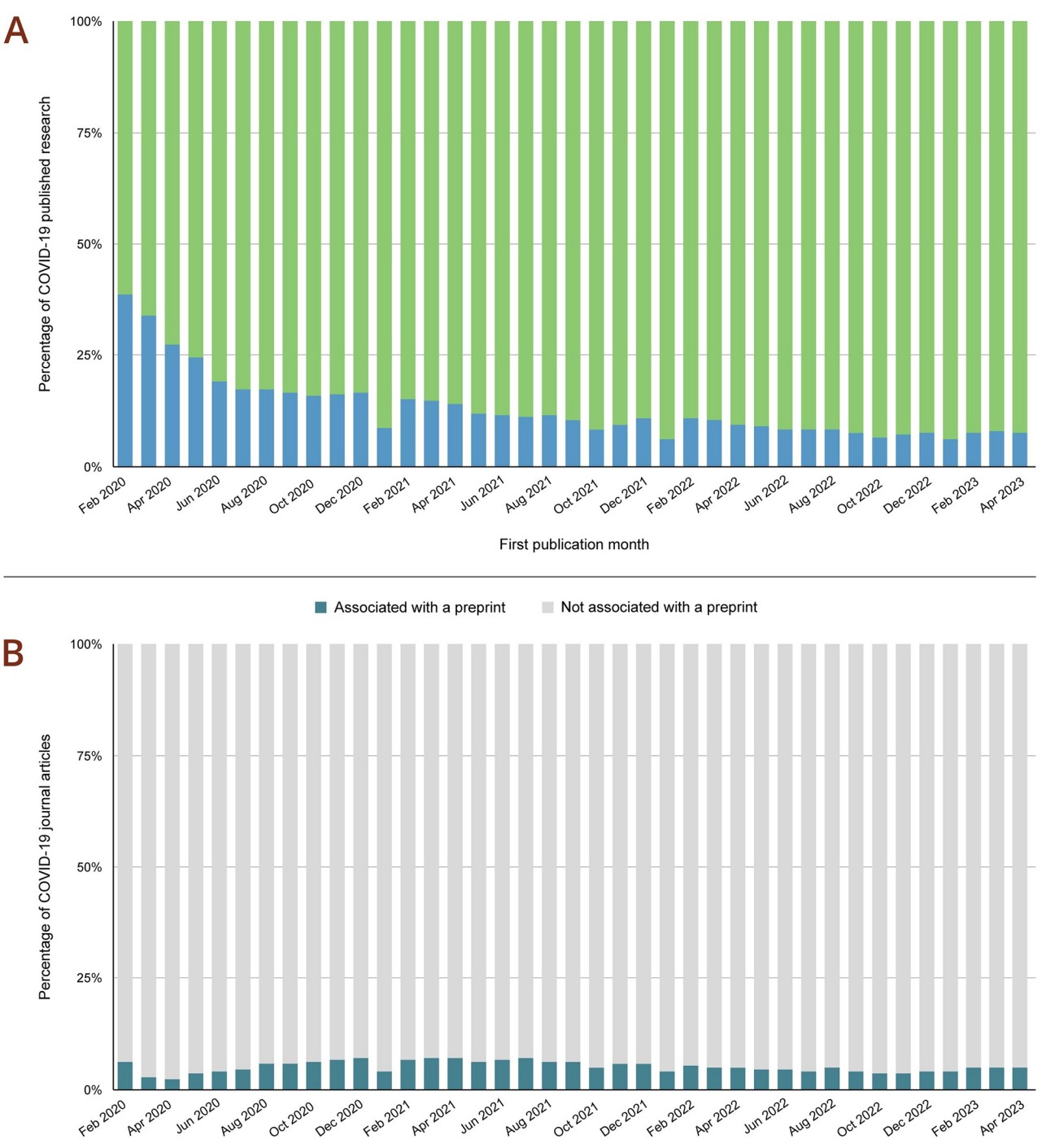

**Fig 2. Preprints comprise a significant portion of COVID-19 research.** (A) Percentage of COVID-19 preprints (in blue) and COVID-19 journal articles (in green) available in Europe PMC by month. This data includes preprints that have been subsequently published as a journal article. (B) Percentage of COVID-19 journal articles preceded by a preprint (in teal). Accessed 8 April 2024.

community governance and sustainability plans for long term availability. This has been exemplified by the discontinuation of Microsoft Academic in 2021 and Meta in 2022 [31, 32]. Lack of transparency in indexing policies and algorithms may also pose a problem. For example, reproducibility in search strategies is key when using such tools for systematic reviews [33]. Several discovery tools offer free options for preprint searches to individual researchers. However, few of them provide free programmatic access and bulk download options. This can pose difficulties for text-mining research involving preprints.

Recognising the importance of preprints in the life sciences and the existing challenges of preprint discovery and reuse, in 2018 Europe PMC started indexing life science preprints alongside peer-reviewed publications [34]. Europe PMC (https://europepmc.org/) is a global open science platform and a life science literature database [35]. Europe PMC is developed by the European Bioinformatics Institute (EMBL-EBI). It is part of the PubMed Central (PMC) International archive network, built in collaboration with the PMC archive in the USA. Europe PMC was originally launched in 2007 as UKPMC—a dedicated archive for full text open access publications supported by UK-based biomedical research funders [36, 37]. In 2012 the database name was changed to Europe PMC due to expansion of its funding consortium and inclusion of funding organisations based in Europe [38]. Currently Europe PMC is supported by a group of 35 international science funders as their repository of choice, providing a Europe PMC plus deposition service (https://plus.europepmc.org/home) for authors supported by these funding organisations.

Europe PMC provides access to millions of abstracts and full text records from a range of sources, including MEDLINE (PubMed), PMC, and Agricola, as well as many preprint servers. All open access content is freely available via the Europe PMC website, Application Programming Interfaces (APIs) and as bulk downloads.

In July 2020 Europe PMC enhanced its preprint coverage by indexing the full text of preprints related to COVID-19. The full text was made available in a standard JATS XML format to support text-mining analyses of the COVID-19 literature [39, 40]. This corpus can be accessed via the Europe PMC website, APIs, and bulk downloads, where licensing allows. This initiative aimed to increase the discoverability of COVID-19-related research and concluded in October 2023. It complemented other projects providing access to COVID-19-related literature [41, 42].

From April 2022 the full text of preprints that acknowledge funding from one or more of the Europe PMC funders from selected servers was also made available in Europe PMC [43]. This initiative aimed to expand the collection of full text preprints for future analyses. Indexing the full text of such preprints also improves visibility of science supported by Europe PMC funders. In a similar effort in January 2023, NIH-affiliated and supported preprints from selected servers became available via PMC USA and PubMed [44].

This paper describes the scope and technical approach to indexing life science preprints in Europe PMC. It summarises our efforts to improve preprint discoverability and help build trust in preprints. We also share lessons learnt in the process.

## Methods

### Technical implementation

**Preprint selection criteria.** Europe PMC includes abstracts and metadata for preprints from preprint servers that fulfil the following criteria (https://europepmc.org/Preprints#preprint-criteria).

- **Scope:** the preprint server should have a significant proportion of life science or interdisciplinary subjects preprints.

- **Volume:** the preprint server should have at least 30 preprints available.

- **Screening:** the preprint server should have a screening procedure detailed in a public statement.

- **Access to metadata:** preprint metadata should be available in machine-readable format, e.g. via an API.

- **Access to full text:** the full text of all preprints should be visible on the preprint server with no barrier to access.

- **Publication ethics:** the preprint server should have a public statement on plagiarism, misconduct, and competing interests policies.

In addition, Europe PMC requires the following metadata in a machine readable format:

- Preprint identifier (currently a Crossref DOI is required)

- Preprint title

- Author names

- Abstract

- Publication date

As of 4 April 2024, Europe PMC indexes preprints from 32 preprint servers (Table 1).

In addition to abstracts and metadata, Europe PMC includes full text of preprints related to COVID-19, as well as preprints supported by the Europe PMC funder consortium.

The COVID-19 preprints were selected from a subset of preprint servers indexed in Europe PMC (Table 1). Coronavirus-related preprints were identified using the following metadata search query in Europe PMC: (TITLE_ABS:"COVID" OR TITLE_ABS:"SARS CoV 2" OR TITLE_ABS:"2019-nCoV" OR TITLE_ABS:coronavirus* OR TITLE_ABS:"corona virus" OR TITLE_ABS:"MERS CoV" OR TITLE_ABS:"Middle East Respiratory Syndrome" OR TITLE_ABS:"Severe Acute Respiratory Syndrome"). In the case of arXiv and SSRN servers, a curated list of COVID-19 preprints was provided by the corresponding server instead. In addition to criteria listed above, contact details for the corresponding author were a prerequisite for indexing the preprint full text in Europe PMC. Preprints and any new versions posted before 1 November 2023 were included, thereafter the initiative ended.

Europe PMC funder preprints are selected from the medRxiv, bioRxiv, and Research Square servers. Those preprints are identified based on the full text funding acknowledgement. The Acknowledgement or Funding section in the PDF file of the full text preprint is text-mined for Europe PMC funder names using a series of Python scripts. Support from at least one of the Europe PMC funders (https://europepmc.org/Funders/) has to be disclosed. Only preprints that have a Creative Commons licence are selected for full text indexing. Preprints and any new versions posted on or after 1 April 2022 are included.

**Preprint ingest and conversion process.** Crossref (https://www.crossref.org/) is a DOI registration agency for scholarly documents, including preprints. Europe PMC uses the Crossref REST API to retrieve new preprint abstracts and metadata as well as updates on a daily basis [45]. Relevant preprint content is identified based on several filters including the DOI prefix and work type. Preprint records added to the Europe PMC database are assigned a unique PPR identifier.

**Table 1. The list of preprint servers indexed in Europe PMC.**

| Preprint server/ platform name | Search syntax in Europe PMC | Preprint server or platform URL | Preprint metadata and abstracts indexed | Preprint full text indexed* |
|---|---|---|---|---|
| Access Microbiology | PUBLISHER:"Access Microbiology" | https://www.microbiologyresearch.org/content/journal/acmi | All preprints | No |
| AfricArXiv | PUBLISHER:"AfricArXiv" | https://osf.io/preprints/africarxiv/ | All preprints | No |
| agriRxiv | PUBLISHER:"agriRxiv" | https://agrirxiv.org/ | All preprints | No |
| AIJR Preprints | PUBLISHER:"AIJR Preprints" | https://preprints.aijr.org/ | All preprints | No |
| ARPHA Preprints | PUBLISHER:"ARPHA Preprints" | https://preprints.arphahub.com/ | All preprints | No |
| arXiv | PUBLISHER:"arXiv" | https://arxiv.org/ | COVID-19 preprints | COVID-19 preprints |
| Authorea Preprints | PUBLISHER:"Authorea Preprints" | https://authorea.com/ | All preprints | COVID-19 preprints |
| Beilstein Archives | PUBLISHER:"Beilstein Archives" | https://www.beilstein-archives.org/xiv/home | All preprints | No |
| BioHackrXiv | PUBLISHER:"BioHackrXiv" | https://biohackrxiv.org/ | All preprints | No |
| bioRxiv | PUBLISHER:"bioRxiv" | https://www.biorxiv.org/ | All preprints | COVID-19 and CC-licensed Europe PMC funder preprints |
| ChemRxiv | PUBLISHER:"ChemRxiv" | https://chemrxiv.org/ | All preprints | COVID-19 preprints |
| EcoEvoRxiv | PUBLISHER:"EcoEvoRxiv" | https://ecoevorxiv.org/ | All preprints | No |
| Emerald Open Research | PUBLISHER:"Emerald Open Res" | https://emeraldopenresearch.com/ | All preprints | No |
| F1000 Research | PUBLISHER:"F1000Res" | https://f1000research.com/ | All preprints | No |
| Gates Open Research | PUBLISHER:"Gates Open Res" | https://gatesopenresearch.org/ | All preprints | No |
| Health Open Research | PUBLISHER:"AMRC Open Res" OR PUBLISHER:"healthopenres" | https://healthopenresearch.org/ | All preprints | No |
| HRB Open Research | PUBLISHER:"HRB Open Res" | https://hrbopenresearch.org/ | All preprints | No |
| MedEdPublish | PUBLISHER:"MedEdPublish" | https://www.mededpublish.org/ | All preprints | No |
| medRxiv | PUBLISHER:"medRxiv" | https://www.medrxiv.org/ | All preprints | COVID-19 and CC-licensed Europe PMC funder preprints |
| NIHR Open Research | PUBLISHER:"NIHR Open Res" | https://openresearch.nihr.ac.uk/ | All preprints | No |
| Open Research Africa | PUBLISHER:"AAS Open Res" OR PUBLISHER:"Open Res Africa" | https://openresearchafrica.org/ | All preprints | No |
| Open Research Europe | PUBLISHER:"Open Res Europe" | https://open-research-europe.ec.europa.eu/ | All preprints | No |
| PaleorXiv | PUBLISHER:"PaleorXiv" | https://osf.io/preprints/paleorxiv/ | All preprints | No |
| PeerJ Preprints | PUBLISHER:"PeerJ Preprints" | https://peerj.com/preprints/ | All preprints | No |
| Preprints.org | PUBLISHER:"Preprints.org" | https://www.preprints.org/ | All preprints | COVID-19 preprints |
| PsyArXiv | PUBLISHER:"PsyArXiv" | https://psyarxiv.com/ | All preprints | COVID-19 preprints |
| Qeios | PUBLISHER:"Qeios" | https://www.qeios.com/ | All preprints | No |
| Research Square | PUBLISHER:"Research Square" | https://www.researchsquare.com | All preprints | COVID-19 and Europe PMC funder preprints |
| SciELO Preprints | PUBLISHER:"SciELO Preprints" | https://preprints.scielo.org/ | All preprints | COVID-19 preprints |
| ScienceOpen Preprints | PUBLISHER:"ScienceOpen Preprints" | https://www.scienceopen.com/collection/5916e67c-0edf-472a-ad8e-6e205a4e080d | All preprints | No |
| SSRN | PUBLISHER:"SSRN" | https://www.ssrn.com/index.cfm/en/ | COVID-19 preprints | COVID-19 preprints |
| Wellcome Open Research | PUBLISHER:"Wellcome Open Res" | https://wellcomeopenresearch.org/ | All preprints | No |

*COVID-19 preprints posted until 1 November 2023. Europe PMC funder preprints posted on or after 1 April 2022.

**Table 2. Methods of PDF/metadata retrieval for full text preprints by server.**

| Preprint server | Method of PDF and metadata retrieval | |
| --- | --- | --- |
| | COVID-19 preprints | Europe PMC funder preprints |
| arXiv | Custom metadata feed + DataCite API | Not applicable |
| Authorea Preprints | Website | Not applicable |
| bioRxiv | bioRxiv API + website | bioRxiv API + website |
| ChemRxiv | ChemRxiv API | Not applicable |
| medRxiv | medRxiv API + website | medRxiv API + website |
| Preprints.org | Crossref API + website | Not applicable |
| PsyArXiv | OSF API | Not applicable |
| Research Square | Research Square API | Research Square API |
| SciELO Preprints | Website | Not applicable |
| SSRN | Custom metadata feed | Not applicable |

To index and display full text for COVID-19 and Europe PMC funder preprints, Europe PMC has set up a processing workflow. The open source Europe PMC plus manuscript submission system is used to process full text preprints [46].

In this workflow PDF documents for each preprint version are obtained from selected preprint servers (Table 2). The files include the manuscript, figures, and supplementary information.

Using the Europe PMC plus submission system, the preprint full text in PDF format and associated files are passed to an external vendor (Molecular Connections, https://molecularconnections.com/) to create the full text JATS XML. Full text preprints are tagged with the article-type attribute "preprint" to differentiate preprints from journal-published articles. After a further quality assurance step by an external vendor to resolve any XML errors, the full text HTML version of the preprint is created via transformation of the JATS XML using an open source XSL stylesheet (https://europepmc.org/xsl/jats2html.xsl).

Full text in JATS XML format for all COVID-19 and Europe PMC funder preprints is indexed for search. Further redistribution is dependent on the permissions of the individual preprint license. This includes display, API access and download options. Preprint license information is identified during metadata retrieval from the preprint server and is captured in the preprint XML during processing. The full text for preprints with a Creative Commons license is automatically released for display and text mining after 14 days. The corresponding author has an option to review the preprint full text during this time. For preprints where no Creative Commons license could be identified, we contact the corresponding author. They are asked to approve the full text, and to add a license to the version of their preprint in Europe PMC that would allow unrestricted reuse for research purposes (https://plus.europepmc.org/user-guide/preprintfaqs#license). If such a license is added, the full text is displayed on the Europe PMC website and made available via the API and bulk downloads as part of the open access subset.

**Handling preprint updates.** *Linking preprints to published journal articles.* Preprints in Europe PMC are linked to the corresponding journal-published versions. The link is established for Europe PMC-indexed journal articles that have a PubMed ID (PMID). For some preprint records the DOI for the journal article is provided by the preprint server in the Crossref metadata. When this is not available, an automated process in Europe PMC links preprints to journal articles based on matching titles and first author surnames. The title matching accounts for changes in text case, whitespaces, and punctuation, while the first author surname is matched based on a similarity threshold. When a version of a preprint is linked to a

corresponding journal article, any previous or subsequent versions that might otherwise not be linked (e.g. due to title or authorship changes) are automatically associated with the same journal article.

*Handling preprint versions*. There are two main ways in which preprint servers indexed in Europe PMC assign persistent identifiers to preprint versions (Table 3):

- Each version of the preprint is assigned its own unique DOI or identifier.

- All versions of the preprint are registered under the same DOI or identifier.

Preprint versions that have a unique DOI or other identifier each receive their own PPRID in Europe PMC. The existence of new versions is detected via Crossref metadata updates or via

**Table 3. Preprint versioning type and version availability in Europe PMC by server.**

| Preprint server or platform name | Versioning type | Preprint version information in Europe PMC |
|---|---|---|
| Access Microbiology | Unique identifier for each version | Available |
| AfricArXiv | Single identifier for all versions | Not available |
| agriRxiv | Single identifier for all versions | Not available |
| AIJR Preprints | Single identifier for all versions | Not available |
| ARPHA Preprints | Single identifier for all versions | Not available |
| arXiv.org | Unique identifier for each version | Available |
| Authorea Preprints | Unique identifier for each version | Available |
| Beilstein Archives | Unique identifier for each version | Available |
| BioHackrXiv | Single identifier for all versions | Not available |
| bioRxiv | Single identifier for all versions | Not available |
| ChemRxiv | Unique identifier for each version and concept identifier for latest version available | Available |
| EcoEvoRxiv | Single identifier for all versions | Not available |
| Emerald Open Research | Unique identifier for each version | Available |
| F1000 Research | Unique identifier for each version | Available |
| Gates Open Research | Unique identifier for each version | Available |
| Health Open Research | Unique identifier for each version | Available |
| HRB Open Research | Unique identifier for each version | Available |
| MedEdPublish | Unique identifier for each version | Available |
| medRxiv | Single identifier for all versions | Not available |
| NIHR Open Research | Unique identifier for each version | Available |
| Open Research Africa | Unique identifier for each version | Available |
| Open Research Europe | Unique identifier for each version | Available |
| PaleorXiv | Single identifier for all versions | Not available |
| PeerJ Preprints | Unique identifier for each version and concept identifier for latest version available | Available |
| Preprints.org | Unique identifier for each version | Available |
| PsyArXiv | Single identifier for all versions | Not available |
| Qeios | Unique identifier for each version | Available |
| Research Square | Unique identifier for each version | Available |
| SciELO Preprints | Single identifier for all versions | Not available |
| ScienceOpen Preprints | Unique identifier for each version | Available |
| SSRN | Single identifier for all versions | Not available |
| Wellcome Open Research | Unique identifier for each version | Available |

specific preprint server APIs. In this case previous versions are linked to the most recent version using an internally developed matching process. When a new version is added, only this latest version will be indexed and available in search results. Previous versions can be accessed through links from the latest version. Preprint versions that share a single DOI or other identifier have a single PPRID in Europe PMC, and only the latest record is available in Europe PMC. No links are established to the previous versions and the version history is not available for such records. The original publication date for the version 1 is retained for such records in Europe PMC.

*Handling preprint withdrawals and removals.* Preprints may be withdrawn or removed from the preprint server due to a variety of reasons [47, 48]. When that happens, typically a withdrawal or removal notice is added to the preprint record as a new version of the preprint. Such notices are often only a few sentences long. Therefore, in Europe PMC those notices are identified based on document length for the full text preprint subset [46]. Withdrawn and removed preprint records are then flagged, manually checked, and tagged with an appropriate article-type attribute in the JATS XML. Once the article-type is correctly tagged, the notice full text is released for display and linking to previous versions on Europe PMC.

In some cases the preprint URL leads to a 404 page, and not even a notification remains. If this is reported to the Europe PMC team the record is removed from Europe PMC as well. Currently there is no automated check to identify deleted preprint records.

**Enriching preprints with relevant resources.**   Along with aggregating preprint content, Europe PMC enriches them in a number of ways. Preprints in Europe PMC are linked to authors, citations, funding, data, reviews, impact metrics, and other resources.

*ORCID iDs.* ORCID iDs are unique researcher identifiers that help disambiguate academic authors. Information about preprint author ORCID iDs is retrieved programmatically from ORCID. Europe PMC also provides an ORCID linking tool (https://europepmc.org/orcid/import). This tool allows authors to connect articles and preprints in Europe PMC to their ORCID records.

*Reviews and evaluations.* Preprints in Europe PMC are linked to preprint evaluations from various preprint review communities (Table 4). Preprint review information is retrieved daily from Crossref, and two preprint peer review aggregation platforms—Sciety (https://sciety.org/) and Early Evidence Base (https://eeb.embo.org/).

The Crossref REST API is queried daily to identify review materials by making use of the 'from-index-date' and 'until-index-date' filters to check for new and updated works in Crossref, combined with filtering for works of type 'peer-review'. DOIs of reviewed preprints are matched to preprint records in Europe PMC.

Preprint review metadata is provided in DocMap format by Sciety and Early Evidence Base and is fetched using the following API calls: https://sciety.org/docmaps/v1/index?updatedAfter=[date] and https://eeb.embo.org/api/v2/docmap/[startdate]/[enddate]/[page] from Sciety and Early Evidence Base, respectively. New and updated DocMap files are converted to XML documents using an open source parser [49] developed by Europe PMC, and DOIs of reviewed preprints are matched to preprint records in Europe PMC.

*Funding information.* Funding information for preprints is retrieved via the Crossref API where available. Mentions of the Europe PMC funder grants are text-mined from the Acknowledgements and Funding sections of full text preprints.

*Biological annotations.* Preprints in Europe PMC are enriched with annotated biological concepts. Annotation types include organisms, data accessions, or experimental methods. Those concepts are identified in abstracts and full text, where available, by the Europe PMC text-mining pipeline [50].

**Table 4. Preprint review groups with review information in Europe PMC.**

| Preprint review provider | Metadata source | Metadata format |
|---|---|---|
| Arcadia Science | Sciety | DocMaps file |
| ASAPbio crowd review | Sciety | DocMaps file |
| Biophysics Colab | Sciety | DocMaps file |
| eLife | Early Evidence Base | DocMaps file |
| EMBO Press | Early Evidence Base | DocMaps file |
| Microbiology Society | Crossref | DOI-associated metadata |
| Life Science Editors | Sciety | DocMaps file |
| Life Science Editors Foundation | Sciety | DocMaps file |
| 2019 Novel Coronavirus Research Compendium (NCRC) | Sciety | DocMaps file |
| NIHR Open Research | Crossref | DOI-associated metadata |
| Peer Community In | Early Evidence Base | DocMaps file |
| PeerJ | Sciety | DocMaps file |
| PeerRef | Early Evidence Base | DocMaps file |
| preLights | Sciety | DocMaps file |
| PREreview | Sciety | DocMaps file |
| Publons | Crossref | DOI-associated metadata |
| Qeios | Crossref | DOI-associated metadata |
| Rapid Reviews Infectious Diseases | Sciety | DocMaps file |
| Review Commons | Early Evidence Base | DocMaps file |
| ScienceOpen | Crossref | DOI-associated metadata |

*External links*. Preprints in Europe PMC are linked to relevant free external resources, for example biological data records or alternative metrics, through the External Links service (https://europepmc.org/LabsLink).

*References*. Where possible, articles within reference lists of the full text preprints are matched and hyperlinked to corresponding records in Europe PMC.

## Data collection and analysis

Data collection and analysis were performed using the R programming language, making use of the europepmc R package [51], which retrieves data from the publicly accessible Europe PMC APIs. A full description of the methodology documented in an R Markdown file and underlying datasets are available in Zenodo [52].

## Results

### Including preprints in Europe PMC search

From the start, it was important for Europe PMC to offer a metasearch solution for life science preprints across many different preprint platforms. As of 4 April 2024, Europe PMC provides access to abstracts and metadata for 761,123 life science preprints from 32 preprint platforms (Table 1).

Since the launch of the Europe PMC project to index preprints in 2018 [34], the number of preprint servers indexed in Europe PMC grew threefold. We continue to monitor the preprint landscape for new platforms that can be added as trusted preprint content sources. It is crucial that preprint content added to Europe PMC conforms to accepted scientific standards. There-fore, we have adopted a set of indexing guidelines for preprint servers (https://europepmc.org/Preprints#preprint-criteria). To be included in Europe PMC, servers must have a public

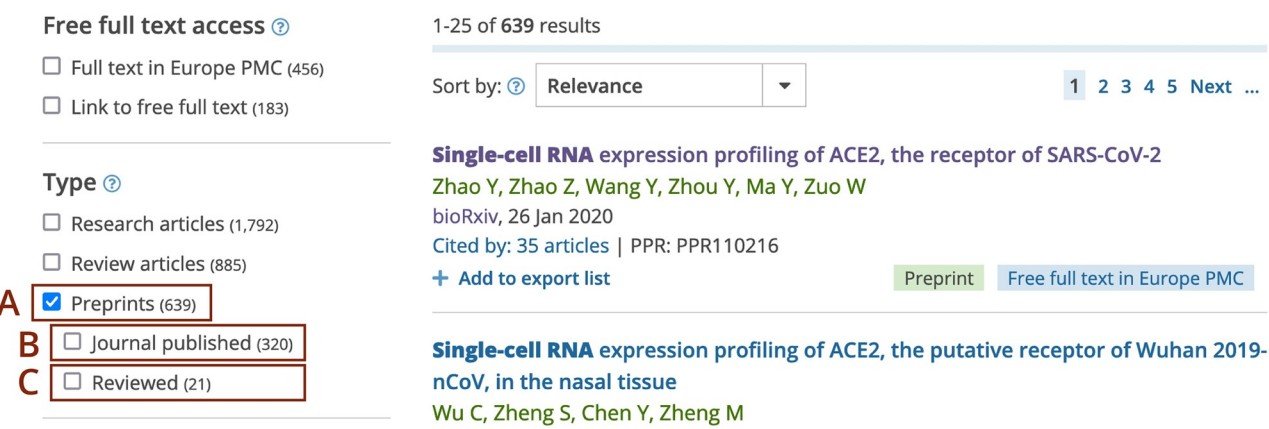

**Fig 3. Europe PMC search filters for preprints, journal-published preprints, and reviewed preprints.** Screenshot of the Europe PMC search results shown for a (*Single-cell RNA expression profiling of ACE2*) query with the Preprints filter on (A). When the 'Preprints' filter is selected, additional sub-filters are displayed: 'Journal published' filter (B) for preprints with an associated journal published version and 'Reviewed' filter (C) for preprints with associated reviews and evaluations.

statement on screening, plagiarism, misconduct, and competing interests. Preprint servers should have no barriers to access the full text for all preprints. This covers subscriptions or sign-in screens. To ensure transparency and improve discoverability, Europe PMC requires a minimal set of metadata. This includes title, abstract, publication date, and author names. Metadata also has to contain a preprint identifier, such as a DOI (digital object identifier). In addition, there are some technical considerations for indexing preprint content. Europe PMC requires access to selected metadata in a machine-readable format. The server has to have a certain volume of preprints posted, with at least 30 preprints available. As a life science database, Europe PMC only includes servers with a significant proportion of life science-related content.

Preprints are included in Europe PMC search results alongside journal publications. If several preprint versions exist in Europe PMC, only the latest version is searchable and shown in search results. Users can access earlier versions through version history on the preprint page (see Table 3 for information on preprint version availability in Europe PMC). When available, the full text of preprints is also included in search. This broadens search results and surfaces preprints that contain search terms not found in the abstracts or metadata. As of 4 April 2024, the full text for 88,831 preprints is searchable in Europe PMC.

Europe PMC uses an open-source Apache Solr search platform (https://solr.apache.org/) that offers full-text faceted search, real-time indexing, and rich document handling. This enables advanced search options for preprints, including filters, as well as Boolean and syntax searches [35, 53]. For example, it is possible to filter the search to return preprints only (Fig 3A). Additional filters help find preprints that have been published as a journal article (Fig 3B), or have an associated review or evaluation (Fig 3C). It is possible to save regular searches and receive email or RSS alerts for new search results. This allows users to keep up with new preprints in the field.

## Tracking changes to preprints

Preprints in Europe PMC are clearly differentiated from peer-reviewed articles. Preprints are labelled with a green 'Preprint' tag, both in search results (Fig 4A), and on the preprint page (Fig 4B). The preprint page also displays a yellow banner indicating to the user that the record

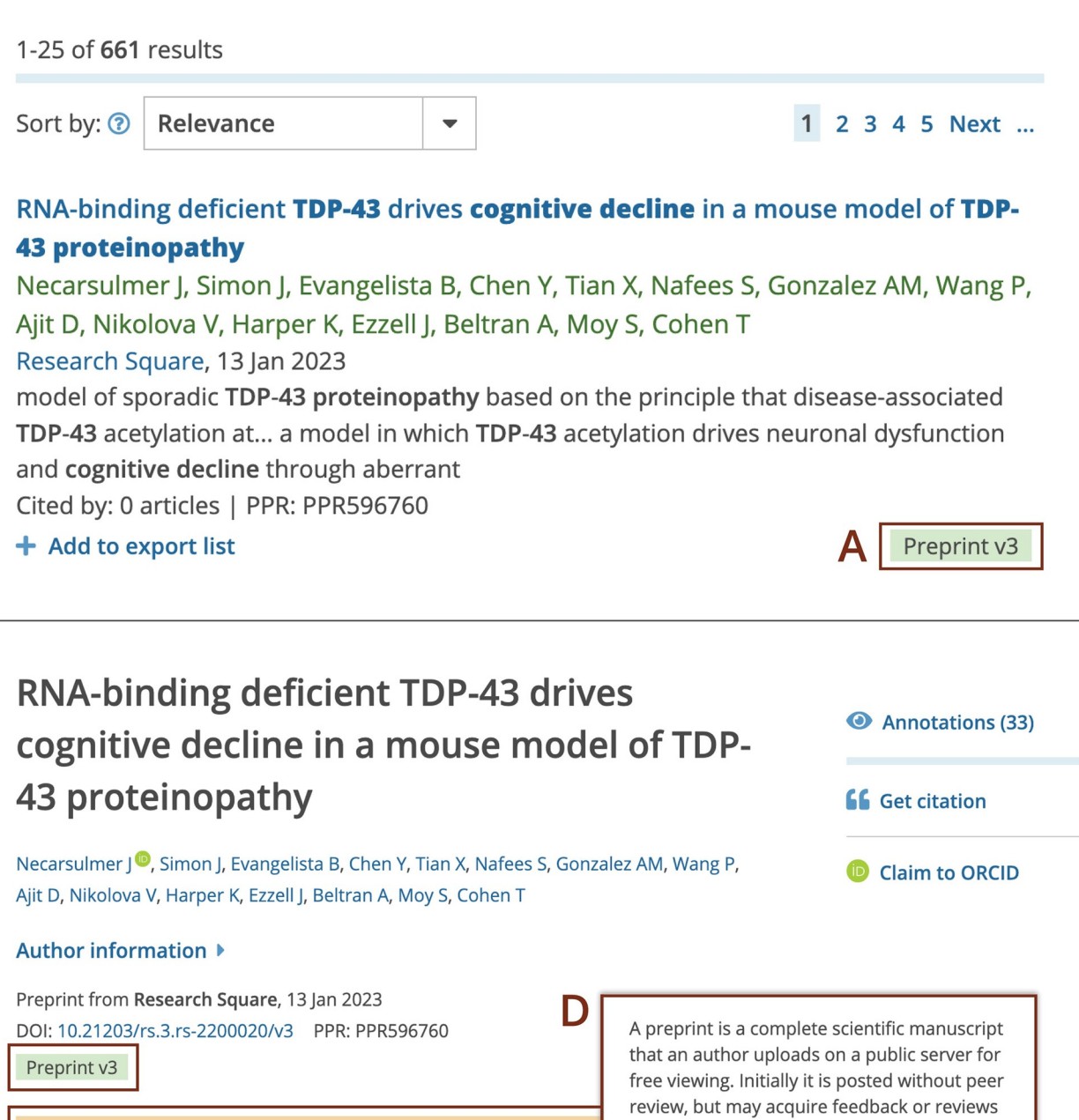

**Fig 4. Display of preprints in search results (top) and on the preprint record page (bottom) in Europe PMC.** The top figure displays a screenshot of the Europe PMC search results page for a (*cognitive decline in TDP-43 proteinopathy*) query. The bottom figure displays a screenshot of the Europe PMC preprint page for the PPR596760 record. (A) Preprints in search results are labelled with a green 'Preprint' tag that also displays version number. (B) The green 'Preprint' tag and (C) a yellow banner indicate that the record being viewed is a preprint. (D) Hovering over the question mark in the banner evokes a pop-up with additional information about preprints. (E) Preprint version history with hyperlinks and publication dates for previous versions.

they are viewing is a preprint (Fig 4C). If available, the preprint review status is also shown in the same banner. Hovering over the question mark sign in the banner brings up additional clarification about preprints (Fig 4D).

An important feature of preprints is that they can be revised by posting a new version. This allows researchers to correct, update, and improve their article. Europe PMC records version information. The version number is indicated in the green 'Preprint' tag on the search results and preprint page. All preprint versions can be accessed from the 'Preprint version history' drop down (Fig 4E). However, version information is only available for some preprints indexed in Europe PMC (Table 3). Preprint servers have different approaches to recording new preprint versions. Some create a unique identifier for each preprint version. Others, notably bioRxiv, medRxiv and SSRN, register a single identifier. In the latter case, the metadata is continuously updated to reflect the most recent version of the preprint. Europe PMC detects new preprint versions through an automated process. This process relies on preprint versions having their own unique persistent identifiers. For preprints with a single identifier, metadata in Europe PMC is overwritten and reflects the latest version. In such cases no version history can be recorded in Europe PMC.

As of 8 April 2024 version information can be retrieved for 423,580 preprints in Europe PMC. Of those, 42,458 (10%) preprints are known to have more than one version. The largest number recorded in Europe PMC for a single record is 64 versions. Version distribution for preprints in Europe PMC is shown in Fig 5.

While some preprints are intended as a final version of record, others are later published in peer-reviewed journals. Where possible, preprints are reciprocally linked to the corresponding journal publications in Europe PMC. The link to the journal article is displayed in the yellow warning banner for all preprint versions (Fig 6A). In turn, the blue information banner on the journal article page provides a link to the latest preprint version (Fig 6B). It is possible to find preprint–journal article pairs in Europe PMC using HAS_PUBLISHED_VERSION:y and HAS_PREPRINT:y searches. As of 4 April 2024, 269,479 of preprints have an associated journal published version in Europe PMC. The share of preprints that have been later published in a journal increases with time. For example, 22% preprints posted in 2023 have been published, in comparison to 62% of preprints posted in 2013 (Fig 6C). The reason for this increase is likely caused by delays associated with the journal publishing process. The actual proportion of published preprints in Europe PMC is likely higher than indicated. If the title or first author changes once the preprint is published, the association between a preprint and the journal article will be missed by the matching algorithm. In addition, preprint–journal article links are currently limited to published versions that have a PubMed ID (PMID) and are indexed in Europe PMC.

Posting a new version of the preprint can allow authors to correct or improve the manuscript. In some cases a preprint may need to be fully suppressed from the preprint server. This can happen due to incorrect data or its interpretation, authorship disputes, legal issues, or as a result of erroneous posting. Preprint platforms have different policies regarding permanence of the preprint content (https://asapbio.org/preprint-servers). There are two main options available to authors: preprint withdrawals and preprint removals. In the case of a withdrawal, the preprint itself is still accessible. It is supplemented with a new version, a withdrawal notice. The withdrawal notice explains that the preprint should not be considered part of the scientific record. This is equivalent to a retraction for journal published articles. In the case of a removal, all preprint versions are deleted and the content is no longer accessible. In some cases, a removal notice replaces the preprint itself [47].

In Europe PMC, removed and withdrawn preprints are clearly labelled with a red warning banner (Fig 7A and 7B). The withdrawal banner displays a link to the preprint server for

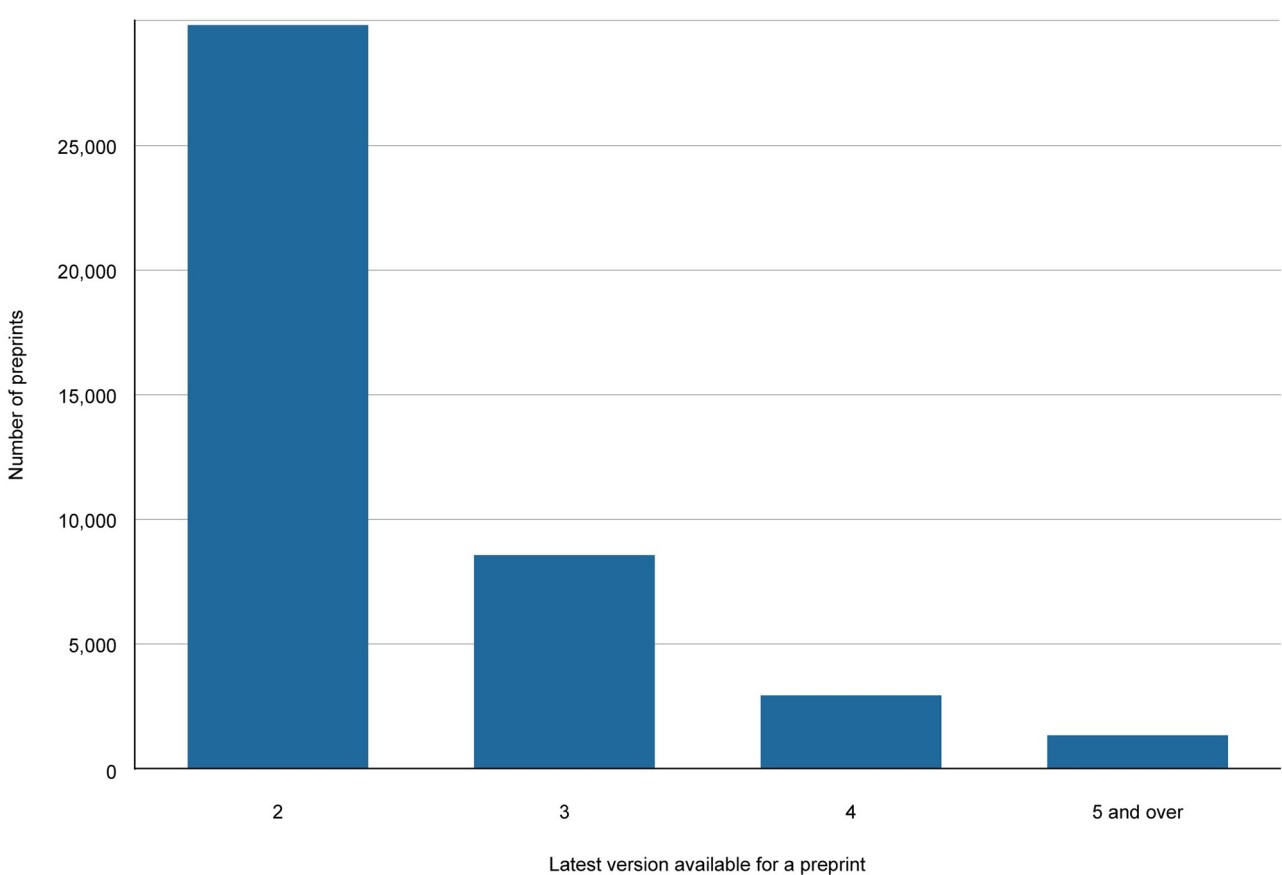

**Fig 5. Number of preprint records that have more than one version in Europe PMC.** This graph displays the number of preprints for which the latest version available is version 2 or higher. The data presented in this graph excludes preprints from servers that issue a single identifier for all versions. Accessed 8 April 2024.

more information. The same banner is also present on earlier versions if those exist in Europe PMC (Fig 7C). Withdrawn and removed preprints in Europe PMC can be found using PUB_TYPE:"preprint-withdrawal" and PUB_TYPE:"preprint-removal" searches, respectively.

Europe PMC determines whether a preprint has been withdrawn or removed using a semi-automated process. This process is a part of the pipeline indexing preprint full text. Withdrawal and removal notices are typically only a few sentences long. Therefore they can be identified based on the length of the preprint full text. Such short preprint records are flagged by the Europe PMC plus system. They are then manually checked and tagged with the appropriate article type. As of 4 April 2024, 56 withdrawn or removed full text preprint records are available in Europe PMC. This comprises 0.1% of all full text preprints. The number does not include preprints that have been deleted from the server entirely, where the preprint URL leads to a 404 page. Preserving the integrity of the scientific record is our main priority as an archive. However, preprint practices differ from those adopted by journal publishers in many aspects, including content permanence. As a content aggregator, Europe PMC reflects the data available via original content source. If a preprint is no longer available and its deletion is reported to the Europe PMC team, the record is fully removed from Europe PMC to reflect the server's approach.

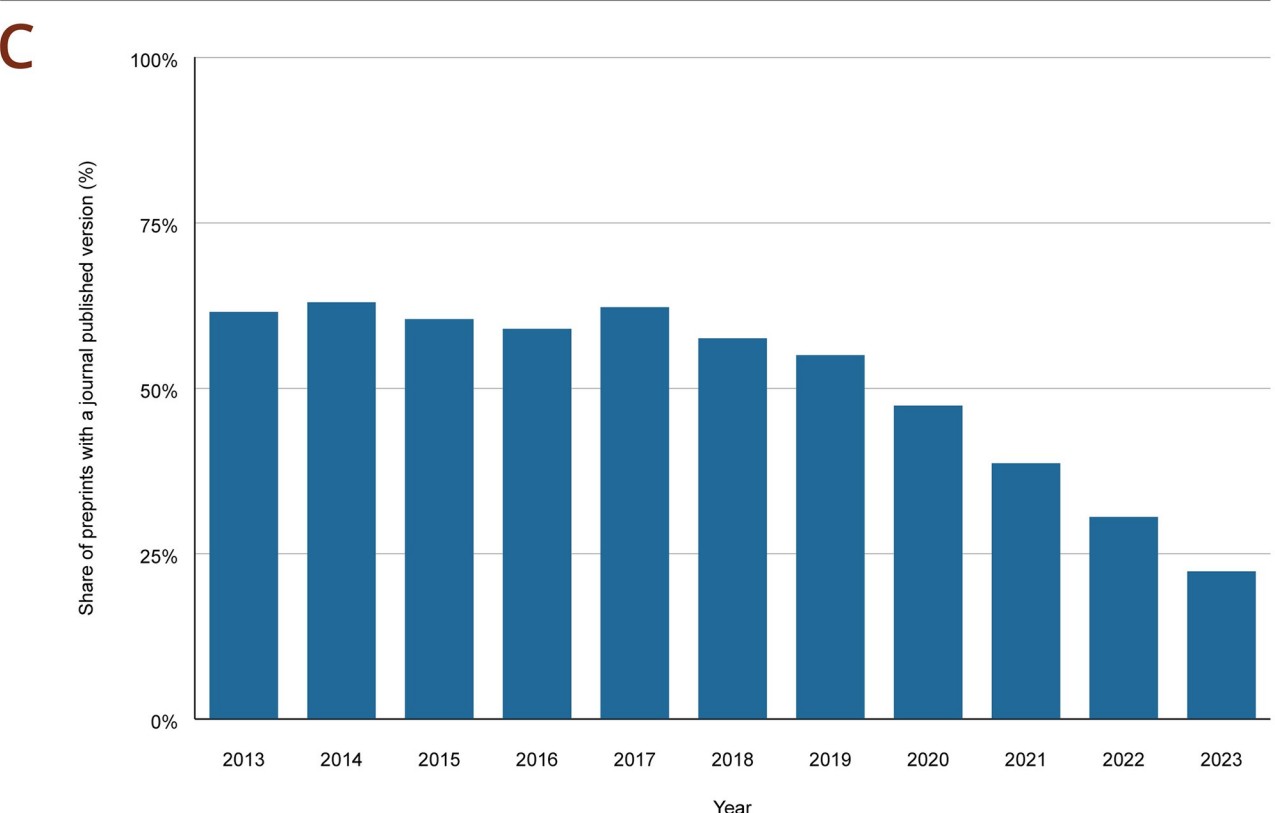

**Fig 6. Preprint–journal article links in Europe PMC.** Screenshots of the information banner on the preprint page with a link to the published journal article (A) and corresponding banner on the journal article page with a link to the latest preprint version (B). A graph showing the percentage of preprints with an associated journal published version in Europe PMC between 2013–2023 (C). The data presented here includes Open Research Platforms, such as F1000 Research, Access Microbiology, etc. Accessed 8 April 2024.

To help users check for updates to preprint records, Europe PMC has developed the Article Status Monitor (http://europepmc.org/ArticleStatusMonitor). This tool identifies most recent preprint versions, journal published versions, withdrawals or removals. Users can submit a preprint DOI or PPRID to view and export updates to the record and access the latest version.

Preprint status changes can be tracked programmatically, with the Status Update Search module of the Articles RESTful API (http://europepmc.org/RestfulWebService#!/Europe32PMC32Articles32RESTful32API/statusUpdateSearch).

**A** ❶ This article is a preprint. This preprint has been removed by the author(s).

**B** ❶ This preprint has been withdrawn by the author(s). View this record on the preprint server for more information.

**C** ❶ This article is a preprint. Version 3 of this preprint has been withdrawn by the author(s).

**Fig 7. Display or removed and withdrawn preprints in Europe PMC.** Warning banners for removed preprint (A), withdrawn preprint (B), and preprint version preceding a withdrawn preprint (C).

### Text-mining and programmatic access for preprints

To support programmatic access to preprints Europe PMC offers freely available APIs. This includes RESTful and SOAP web services with output in XML, JSON or Dublin Core formats. Preprint metadata from preprint servers is enriched with data from other sources. The core API response includes information about funding, reviews, author ORCID iDs, or journal published versions for preprints in Europe PMC. The search syntax reference (https://europepmc.org/searchsyntax) can be used to create advanced programmatic preprint queries, for example to find refereed preprints supported by a particular funding agency.

Preprint abstracts as well as full text, open access preprints, can be downloaded in bulk on the Europe PMC FTP site. Full text and metadata is shared in JATS XML format. This provides a unique set of documents for text-mining and machine-learning purposes.

Europe PMC uses text mining to surface evidence presented in preprints. This process annotates preprints with relevant biological concepts. Example annotations include experimental methods, organisms, diseases, or data accessions. Europe PMC identifies these concepts using text-mining algorithms [37]. The text-mining process covers preprint abstracts and open access full text. Annotations can be accessed using the freely available Europe PMC Annotations API (https://europepmc.org/AnnotationsApi). Annotations can also be highlighted in the text using the SciLite tool [50]. This is possible for all preprint abstracts, and full text preprints with a license that allows derivatives of the work. This is restricted to full text preprints with a CC-BY, CC-BY-NC or CC0 license. Individual annotations are linked to records in public databases or controlled ontologies. For example, a gene ontology term found in the text will be linked to a corresponding record in the Gene Ontology database [54]. This helps foster connections between scientific literature and data.

### Incorporating preprints into Europe PMC infrastructure

Europe PMC aims to establish preprints as first class research outputs. To this end preprints are linked to relevant scholarly resources. All of the relevant content is summarised on the preprint page. This includes the preprint abstract and full text where available (Fig 8A). Europe

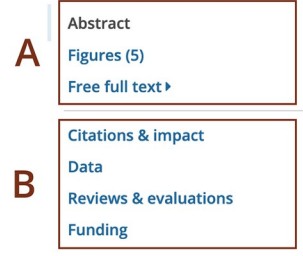

Non-neuronal expression of SARS-CoV-2 entry genes in the olfactory system suggests mechanisms underlying COVID-19-associated anosmia

Brann DH [1], Tsukahara T [1], Weinreb C [1], Lipovsek M [2], Van den Berge K [3], Gong B [4], Chance R [5], Macaulay IC [6], Chou H [5], Fletcher R [5], Das D [5], Street K [7], de Bezieux HR [4], Choi Y [8], Risso D [9], Dudoit S [3,4], Purdom E [4], Mill JS [10], Hachem RA [11], Matsunami H [12] … [Show all 25] … Datta SR [1]

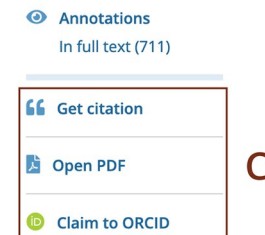

**Fig 8. Preprint record display in Europe PMC.** Screenshot of the preprint page for the PPR130125 record. (A) Preprint abstract, as well as figures and full text, where available, are accessible from the navigation panel on the left. The same panel provides access to additional sections with links to related content (B), such as 'Citations & impact', 'Data', 'Reviews and evaluations', and 'Funding'. The navigation panel on the right provides options to 'Get citation', 'Open PDF', or 'Claim to ORCID' (C).

PMC also provides tools to incorporate preprints into research workflows. Users can cite a preprint using the 'Get citation' feature. Preprint authors can add the preprint to their ORCID profile using the 'Claim to ORCID' option. Those tools are available within the navigation panel on the right (Fig 8B). The navigation panel on the left (Fig 8C) provides access to related content. This covers data behind the paper, peer review materials, or impact metrics. These resources are available within corresponding sections on the preprint page.

The 'Citations and impact' section displays traditional citations and alternative metrics. The number of supporting or contrasting citations available via the 'scite' tool [55] is shown where possible. Users can also find data citations to the preprint, and the number of preprint recommendations by various groups in this section.

The 'Data' section summarises all research data associated with the preprint. It lists data citations, either as data accessions or data DOIs for over 60 different life science databases [56]. Europe PMC identifies citations of data in the text of the preprint using text-mining. Those citations are then linked to the corresponding data record in the external database. The 'Data' section also displays citations from database records to the preprint. For example, if a preprint has been curated by a knowledgebase, or if data behind the preprint was deposited to a dedicated archive.

The 'Funding' section contains funding information for preprints in Europe PMC. Where available, grant details include funder name and grant identifier. Additional information is displayed for grants awarded by Europe PMC funders (https://europepmc.org/Funders/). This covers grant award title, principal investigator name and affiliation. Data provided to Europe PMC directly by the Europe PMC funders is freely accessible via the Grant Finder tool (https://europepmc.org/grantfinder).

## Linking preprints to reviews and evaluations

The process of posting preprints is decoupled from journal-organised peer review. This allows various preprint review initiatives to experiment with new forms of feedback. As the number of such initiatives grows, it becomes increasingly important to connect preprints with existing reviews that are often scattered across different platforms [57].

For some time Europe PMC has linked peer reviews to preprints via the external links scheme (https://europepmc.org/LabsLink). This approach enabled readers to easily find preprint evaluations but provided limited context, lacking information about review type, date, or name of the reviewer.

To enrich preprint review information in Europe PMC we collaborated with two preprint review aggregation platforms, Sciety and Early Evidence Base. To represent different review systems these platforms have adopted the DocMaps framework [58]. DocMaps is a community-endorsed open-source project led by Knowledge Futures. It uses a standard vocabulary to record steps in the review process. By incorporating DocMaps into Europe PMC a more comprehensive display of preprint review activity could be shown. Some preprint evaluation groups are not yet included in Sciety or Early Evidence Base. Therefore we created a pipeline to pull preprint review metadata available in Crossref. Altogether Europe PMC collates review information from 20 platforms and communities (Table 4). As of 4 April 2024 there are 12,209 reviewed preprints in Europe PMC, which can be found using the (HAS_VERSION_EVA-LUATIONS:Y) search syntax.

Preprint review status is disclosed in the yellow warning banner (Fig 9A and 9B). For earlier preprint versions it is also shown in the version history (Fig 9C). The banner provides a link to the 'Reviews and evaluations' section on the preprint page. This section collates review information available in Europe PMC. Readers can explore the review timeline with review activity grouped by platform. For each review Europe PMC provides a link to the full assessment. Where available, Europe PMC shows evaluation date, review title or type, and evaluator name (Fig 9D).

## Discussion

### Benefits of including preprints in Europe PMC

Europe PMC started indexing preprints in 2018. At that time our main goal was to make preprints more discoverable and enable their inclusion into research workflows [34]. Since then, further benefits of indexing preprints in Europe PMC became apparent. This includes opportunities for meta research as well as new publishing and review models.

**Accelerating discovery of preprints.** Europe PMC offers a single search for life science journal articles and preprints. This improves preprint visibility, aided by preprints' default inclusion in search results. Indexing the full text of preprints also increases discoverability. Europe PMC offers full text search for the COVID-19 and Europe PMC funder subsets. This helps surface preprints with search terms found beyond the preprint title or abstract.

Continued proliferation of preprint servers can make it difficult to collate all relevant research for literature reviews, as users may not be aware of emerging preprint platforms. Europe PMC enables comprehensive reviews of preprinted literature. Currently Europe PMC aggregates preprints from over 30 different servers. We continue to increase preprint coverage through a regular monitoring process and welcome inclusion requests directly from preprint servers. To support users who want to follow new preprints in their field, Europe PMC provides saved search and email alert features [53].

**Supporting innovative publishing and peer review models.** Preprints provide a way to experiment with new forms of communication and innovative review models. Currently, there is a great variety of publishing workflows that incorporate preprints. Journals may offer concurrent submission, use journal-agnostic review services, or opt for post-publication review [59, 60]. Diversity in this space provides an opportunity to redefine scholarly communication, but it can be difficult to adapt existing systems to new approaches.

Europe PMC integrates preprints with associated versions, reviews, and published journal articles. This brings together content across different platforms and provides a complete picture of the publishing process. Surfacing preprints and preprint reviews helps recognise and credit preprint authors and referees. Europe PMC works closely with the preprint community to better understand their needs [47]. This allows us to build more flexible tools that accommodate

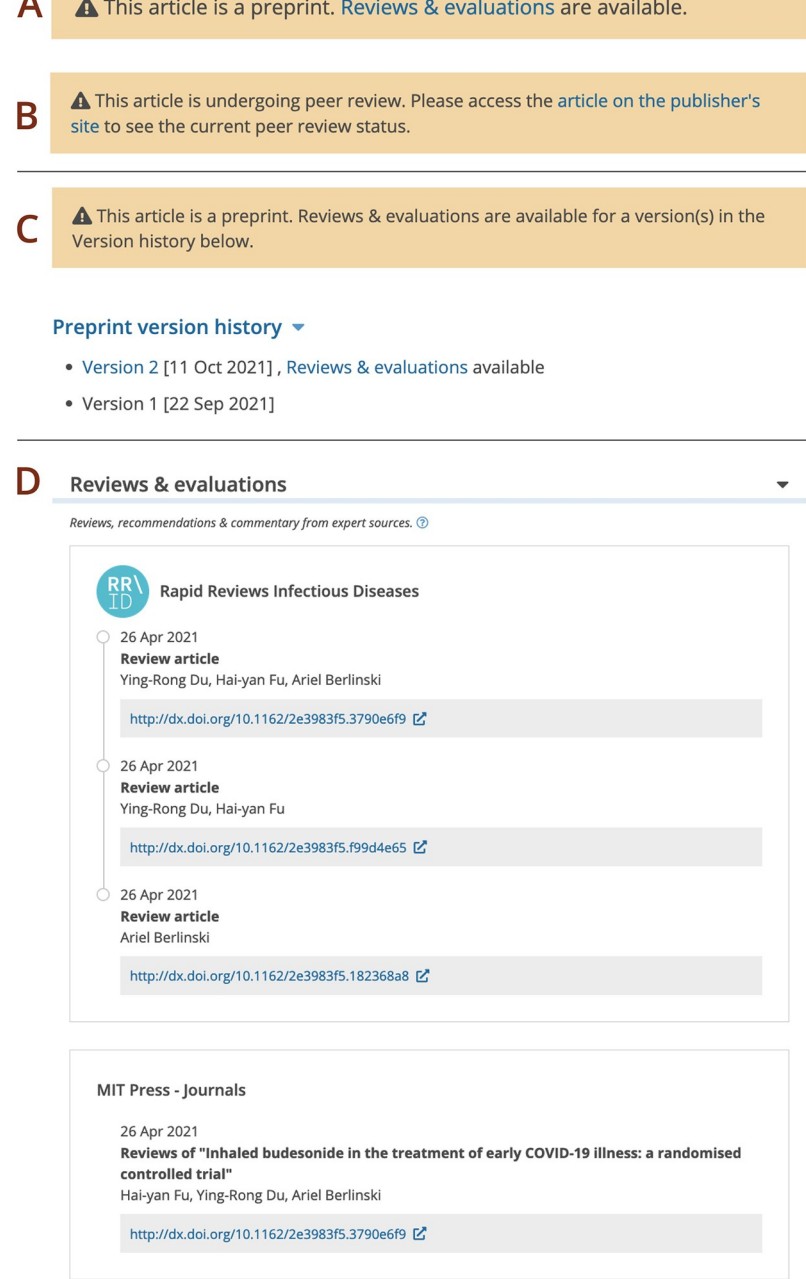

**Fig 9. Linking reviews to preprints in Europe PMC.** Information banner for a reviewed preprint (A) and a preprint posted on an Open Research platform (B). (C) Information banner and version history indicate that reviews and evaluations are available for a later version of a preprint. (D) Screenshot of the 'Reviews and evaluations' section for the PPR279548 record.

evolving practices. For example, Europe PMC displays modified banners for preprints from Open Research Platforms (Fig 9B). Those platforms coordinate peer review immediately on submission and therefore do not fit the standard definition of a preprint server.

**Building trust in preprints.** Since preprints lack the traditional endorsement provided by journal peer review, there is a growing need for the scientific community to find new ways to

evaluate preprinted research. Indexing preprints in Europe PMC helps build trust and support continued preprint adoption in many different ways.

Collating information about available reviews and recommendations on the preprint page helps readers to assess findings reported in preprints. Users that want to limit their search to preprints that have been evaluated by experts can do so using a dedicated filter.

Adopting indexing criteria for preprint servers, such as a requirement for a public statement on screening, plagiarism, and misconduct policies, ensures that preprints in Europe PMC conform to accepted scientific standards.

Comprehensive, high quality metadata can improve transparency and provide credibility for preprints. Preprints in Europe PMC are clearly labelled and distinguishable from journal articles. Version information is displayed where available and can be easily accessed from the 'Preprint version history' menu. Preprints are also reciprocally linked to the corresponding peer reviewed publications.

As concerns regarding preprints' permanence can erode public trust in science, Europe PMC follows best practices for handling preprint withdrawals and removals. In addition to providing clear notices for affected records, Europe PMC offers a way to filter for withdrawn and removed preprints by using dedicated search syntax. Users can also get notifications in the case of withdrawal or removal of a particular record through email alerts or Article Status Monitor tool. To support other tools and services that track preprint updates, for example citation managers, information about preprint withdrawals and removals can also be retrieved programmatically using the Article Status API (https://europepmc.org/RestfulWebService#!/ Europe32PMC32Articles32RESTful32API/statusUpdateSearch).

Finally, access to underlying data, information on authors, affiliations, and funding can serve as indicators of trust for preprints. Linking preprints in Europe PMC to other research outputs provides a way to assess the evidence presented in preprints in the absence of journal peer review.

**Integrating preprints into research workflows.**   The inclusion of preprints in Europe PMC is an important step in establishing preprints as first class research outputs and integrating preprints into the scholarly infrastructure. Europe PMC tools enable inclusion of preprints into research workflows, such as article citation, funding reporting, and credit and attribution.

When working with preprints, researchers need to know which version to cite, what changes were made to the published version, and whether conclusions still stand after a withdrawal [47]. The Europe PMC Article Status Monitor tool helps users to check for various updates to preprint records, while Europe PMC 'Get citation' feature makes it easy to cite preprints or add them to a reference manager tool.

At present, several research funding organisations take preprints into consideration in grant award decisions [19], and many preprints provide information on funding and support. Europe PMC collates funding information for preprints in a dedicated 'Funding' section on the preprint page. Europe PMC also enables funders to monitor open data compliance associated with their funding by linking preprints to underlying data, such as data DOIs and accessions text-mined from preprint records.

**Supporting preprints inclusion in meta-analyses and systematic reviews.**   With continued adoption of preprints in the life sciences they are increasingly included in meta-analyses of research literature [61]. Systematic evidence searches require comprehensiveness, transparency and reproducibility [62]. Europe PMC enables preprint inclusion in evidence synthesis through comprehensive coverage of biomedical preprints across multiple platforms, extensive support documentation, and advanced search functionalities.

As a long-standing public service provider that has adopted the Principles of Open Scholarly Infrastructure (POSI) [63], Europe PMC is committed to long term preservation of

scholarly records, sustainability, and open access to research literature. This is important for systematic reviewers that rely on lasting availability of content, and transparency in indexing policies and algorithms for reproducible search strategies.

Europe PMC users can create complex, customised queries for preprints using Boolean operators, syntax for field searches, as well as advanced search options. Providing a full text search for preprints in Europe PMC increases recall and allows reviewers to identify specific study types, or limit their search to specific preprint sections, such as Figures, Results, or Methods. Systematic reviewers can export all or a selection of search results in various data formats including XML and CSV.

As a result, Europe PMC adoption as a search engine for preprints in systematic reviews of the research literature has been referenced by numerous studies [64–73].

**Providing programmatic access for preprints.**   While expert meta-analyses are crucial to keep up with the newly published evidence, they are often reliant on several manual, resource-intensive steps [74]. Automated information extraction from a large bibliographic database requires access to programmable interfaces.

For users carrying out bioinformatic studies or literature reviews, Europe PMC offers a single API and a bulk download option for free and open access to preprint abstracts and appropriately licensed full text preprints from many different servers. This makes analysis easier and accelerates research of the preprint literature itself. It also enables creation of new tools and cross-platform integrations. For example, the Europe PMC API powers FlyBase preprint search for preprints relevant to the *Drosophila* community [75] and StemRxiv preprint search from the StemJournal for preprints on stem cell research (https://stemjnl.org/preprint-search). Sciety uses the Europe PMC Articles API to retrieve preprint abstracts and display them alongside reviews and evaluations [76].

Crucially, Europe PMC offers one of the largest corpora of preprinted literature in a standard, machine-readable format, providing a unique set of documents for text-mining and machine-learning purposes. In addition, preprints in Europe PMC are incorporated into the knowledge graph and enriched with additional data, including links to journal publications, funding information, etc. High quality open metadata associated with preprints in Europe PMC supports analysis of preprint practices, for example around the impact of peer review, sharing behaviours, the availability of supporting data for preprints, ORCID uptake and so on, at a scale and breadth not previously possible. At present, the full text preprint subset available in Europe PMC has been used for various analyses of research reported in preprints [17, 77–81].

While preprints offer free access for readers, they are not always free to reuse. License limitations can make it difficult to include preprints in large scale analyses. Clearly stating the licence conditions is vital to ensure that the work can be reused appropriately. In some cases license information may be difficult to retrieve programmatically. Europe PMC engaged with preprint servers to ensure that a preprint's license is shared as a part of the preprint metadata in Crossref, supporting reuse and maximising the impact of preprinted research. Our outreach efforts significantly increased the number of preprints with license information both in Crossref and in Europe PMC, adding license metadata for preprints from ARPHA Preprints, Qeios, bioRxiv and medRxiv. As Crossref is used by many other discovery services, we hope that this enables wider redistribution of preprints and increases their visibility.

## Limitations and technical challenges of indexing preprints

Fully realising the potential of preprints by the scholarly community requires a great degree of coordination between servers [47, 82]. Although there are several best practice

recommendations jointly developed by the preprint community [47, 83], no standards are currently enforced for preprints. As a preprint aggregator, Europe PMC has faced several technical challenges related to preprint management and variability in metadata standards that exist between preprint platforms.

**Distributed access points.** A distributed network of preprint servers ensures a degree of autonomy for each preprint community. However, it also creates a challenge of retrieving preprint metadata from different platforms for literature databases, such as Europe PMC. Currently Europe PMC relies on the Crossref API to retrieve preprint metadata and updates to preprint records. Many of the preprint servers that include life science content register DOIs and deposit associated metadata for preprints in Crossref. However, some, for example arXiv (https://arxiv.org/), Jxiv (https://jxiv.jst.go.jp/index.php/jxiv), and ChinaXiv (https://chinaxiv.org/home.htm), use other DOI registration agencies, and some, such as RINarxiv (https://rinarxiv.brin.go.id/lipi), do not register DOIs at all. This does not constitute an outright barrier to inclusion in Europe PMC, but requires free API access and documentation provided by the server, as well as additional resources from the indexer side to set up separate ingest processes for new content sources.

**Limited metadata.** Europe PMC requires a minimal set of metadata as part of its preprint indexing guidelines to ensure discoverability of preprints in search results. This includes a persistent identifier, title, abstract, posted date, and author names. However, there is a lot of variability in the scope of metadata deposited by preprint servers to Crossref, and some preprint servers do not supply all of the metadata fields that Europe PMC requires. This prevents their inclusion in Europe PMC.

**Divergence of practices and standards.** As the preprint ecosystem evolves, the importance of community standards and best practices for preprints becomes more apparent. Aggregating preprints from different sources poses a challenge of accurately representing different approaches. For example, the latest preprint version is equivalent to the peer reviewed article for Open Research platforms and journals adopting the publish-review-curate model [17, 21–24]. Distinguishing these records requires adoption of a basic taxonomy for document versions. Harmonising on these issues will help build a more transparent system.

**Lack of machine-readable status updates.** Unlike journal articles, preprints are not static in their nature and can often be revised, reviewed, published, withdrawn, or removed. Tracking these changes can be difficult due to a number of reasons.

Handling of preprint versions differs from server to server. Some best practice recommendations call for a unique identifier for each preprint version [47], while Crossref recommends updating the preprint metadata using relations tags for new versions (https://www.crossref.org/documentation/research-nexus/posted-content-includes-preprints/). Where these guidelines are not followed, version history can not be recorded in Europe PMC. Gaps in Crossref metadata deposition lead to missing links between preprints and published articles [84]. While basic automated mechanisms can establish those connections without a metadata flag, they may fail if authors or the title change from preprint to the journal version. Finally, preprint withdrawal or removal statuses are not currently available from a centralised source in a standard machine-readable format. This prevents Europe PMC and other providers from automating updates to the preprint record and requires laborious manual checks.

**Text-mining barriers.** Text-mining requires access to full text in a structured machine-readable format. Preprints are often submitted by the authors as PDF files. Thus there is a need to convert the full text of preprints to a suitable format, such as XML. Regrettably, conversion costs often make it prohibitively expensive.

Europe PMC is able to create a full text preprints corpus in XML format thanks to support from its funders. Our first full text preprint project focused on COVID-19 preprints [39]. This

initiative was supported by a joint award from Wellcome, UK Medical Research Council (MRC), Swiss National Science Foundation (SNSF) and endorsed by the Chief Scientist of the World Health Organization (WHO). The full text for all relevant COVID-19 preprints was converted to XML irrespective of the preprint licence. For preprints with a Creative Commons license or similar the full text could then be displayed on the Europe PMC website for reading. The preprint full text was also included in the open access subset for bulk downloads and distributed via Europe PMC APIs. Where the current preprint license did not allow reuse, the author was asked to apply the Europe PMC open access licence. The lack of author licence approval resulted in an excess of 27,000 COVID-19 preprints with full text searchable in Europe PMC but not available to read and text mine. This hinders reuse of the full text preprint corpus and reduces its benefits. As a result, when the Europe PMC funder preprint initiative was launched only preprints with Creative Commons licences were selected for inclusion. Due to associated conversion costs this initiative is currently limited to preprints that acknowledge funding from Europe PMC funders.

Some servers create XML files for preprints shared on their platform, this includes bioRxiv and medRxiv. However, in some cases the license selected by the author prevents redistribution, precluding Europe PMC from utilising the XML for display on Europe PMC.

## Future of preprints in Europe PMC

**Increasing preprint coverage.**   When preprints were first included in Europe PMC in 2018, this was limited to servers that registered Crossref DOIs. Using Crossref provided Europe PMC with a robust centralised process to pull relevant preprint metadata in a standard format. While the majority of the life science preprint servers are current Crossref members, some other servers use alternative DOI registration agencies, including DataCite (e.g. arXiv) or Japan Link Center (e.g. Jxiv). In the future we hope to extend the Europe PMC preprint pipeline to be able to add preprint metadata from sources other than Crossref.

**Enriching preprint metadata.**   Preprint servers often operate under resource constraints, which can make it difficult to capture comprehensive metadata for every record. One example of limited Crossref metadata for preprints is lack of author affiliations, which are considered desired but not essential metadata by the current guidelines for preprint servers [47].

As Europe PMC already generates XML files for full text preprints we can extract additional information that is not available in Crossref. This allows us to surface affiliations for full text preprints in Europe PMC. As a potential next step we plan to add Research Organization Registry identifiers (ROR IDs) to affiliations for Europe PMC-generated full text preprints. ROR provides open persistent identifiers for research organisations that help disambiguate institutional names and connect them to relevant research outputs.

Another example of the preprint metadata gap concerns preprint–journal article relationships. Crossref estimates that around 50% of existing preprint–journal article links are not available in the metadata deposited by its members [85]. While Europe PMC supplements Crossref metadata with an in-house matching algorithm, no association is currently established for preprint–journal article pairs where the title or first author has changed, or the published version lacks a PubMed ID. To mitigate this in the future, we plan to retrieve all preprint–journal article links from Crossref, including those where the published version is not indexed in Europe PMC. In 2023 Crossref employed an automated matching strategy to identify the missing preprint–journal article relationships [86]. Once this data is available through a stable public API, it will be made discoverable in Europe PMC as well.

**Including preprints in the citation network.**   Citation information in Europe PMC is currently retrieved from PubMed and is limited to journal articles. In the future we plan to

extend the Europe PMC open citation network to include preprint citations. Potential citation sources include openly available Crossref metadata supplied by publishers, as well as citations determined from the reference lists of full text publications and preprints in Europe PMC.

**Expanding links between preprints and reviews.**   While DocMaps provide a comprehensive schema for capturing preprint review information, currently they are only created for a limited number of review groups. Some platforms that offer preprint review instead choose to register DataCite DOIs for referee reports, for example F1000 Open Research Platforms. To ensure comprehensive coverage of preprint evaluations we will investigate options for capturing review information currently missing from Europe PMC using the DataCite API (https://api.datacite.org).

**Standards development.**   Europe PMC has played a leading role in preprint community standards development and helped actualize recommendations for metadata practices through a series of stakeholder events [47, 87].

In the future we will continue to work closely with the preprints community on standards for preprint metadata and full text content. Standards for notifications for preprint withdrawals and removals, as well as other metadata events, such as the linking of reviews and evaluations comments to a preprint, are of particular relevance to our work.

## Acknowledgments

We thank arXiv and SSRN staff for providing us with a curated list of COVID-19 preprints during the COVID-19 full text preprint initiative.

## Author Contributions

**Conceptualization:** Mariia Levchenko, Johanna McEntyre, Melissa Harrison.

**Data curation:** Michael Parkin.

**Formal analysis:** Michael Parkin.

**Funding acquisition:** Johanna McEntyre, Melissa Harrison.

**Software:** Michael Parkin.

**Supervision:** Melissa Harrison.

**Visualization:** Mariia Levchenko.

**Writing – original draft:** Mariia Levchenko.

**Writing – review & editing:** Michael Parkin, Johanna McEntyre, Melissa Harrison.

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
