## [Decision Letter · Decision Letter 0]

3 Jun 2024

PONE-D-24-15144Enabling preprint discovery, evaluation, and analysis with Europe PMCPLOS ONE

Dear Dr. Levchenko,

Thank you for submitting your manuscript to PLOS ONE. After careful consideration, we feel that it has merit but does not fully meet PLOS ONE’s publication criteria as it currently stands. Therefore, we invite you to submit a revised version of the manuscript that addresses the points raised during the review process. First of all **I want to thank the 2 reviewers** for their very useful comments that helped me to reach a decision. I choose major revisions in order to account for the opinion of reviewer 2, but I also acknowledge that most of the comments seems pretty easy to address. Therefore I'm quite confident that it will be easy for you to answer. I have a few additional "editorial" comments:- please make sure that you indicate wether your protocol was pre-registered or not. If not, please give an explanation and add some words in the limitation section.- please add a few words about limitations in the abstract and add a specific "limitations" section that summarize the main limitations of your work. This is in order to avoid any spin ; - please select the most appropriate guideline from the equator network (I agree that there is no guideline that has a perfect fit with your research but you may want to choose the closest one) ;  

We look forward to receiving your revised manuscript.

Kind regards,

Florian Naudet, M.D., M.P.H., Ph.D.

Academic Editor

PLOS ONE

Reviewers' comments:

Reviewer's Responses to Questions

**Comments to the Author**

1. Is the manuscript technically sound, and do the data support the conclusions?

Reviewer #1: Yes

Reviewer #2: Yes

2. Has the statistical analysis been performed appropriately and rigorously? 

Reviewer #1: Yes

Reviewer #2: N/A

3. Have the authors made all data underlying the findings in their manuscript fully available?

Reviewer #1: Yes

Reviewer #2: Yes

4. Is the manuscript presented in an intelligible fashion and written in standard English?

Reviewer #1: Yes

Reviewer #2: Yes

5. Review Comments to the Author

Reviewer #1: In this important report, the authors share their experiences with indexing life science preprints, in an effort to promote the discoverability and acceptance of preprints. The authors comprehensively describe how they automated the ingestion of preprints to the Europe PMC database and its enrichment with metadata, including full texts available for some special collections related to COVID-19 or supported by Europe PMC funders. The manuscript includes useful search strategies that can assist researchers in their queries, including programmatic access via different APIs. The authors make a compelling case for a central preprint database that combines data from major preprint servers in a standardized manner, despite the technical difficulties associated with this endeavor.

Minor points:

Lines 71-76: It was unclear from the description and analysis code whether the total number of articles in PubMed includes preprints or not. If it does, then it is appropriate to refer to the share and percentage of preprints. If not, then the measure is preprints per article and not a percentage, nor a share. Please clarify if the number of preprints is included in the total. Incidentally, when I ran the code, the result was exactly 12 and not over 12, but this is probably due to updates in the database.

Figure 2 and lines 90-91: The numbers were higher than 30 only for the first two months, so it would be more precise to replace “few” with “two”. Using the code provided on zenodo, it seems that between 2 and 6% of the journal articles were associated with a preprint (hasPreprint == TRUE). It would be nice to reflect this in this figure, by splitting the articles into articles associated with a preprint or not. This way the conversion of preprints to journal articles becomes more apparent and complements the analysis in Fig. 6.

Line 112: From a security and privacy standpoint it is best to avoid using link shorteners, as they obscure the domain of the landing URL. Please replace with the original URL instead.

Lines 206-210: For better methods reproducibility it would be good to know which tools and external vendors are used for the conversion of PDF to JATS XML and to HTML.

Lines 230-231: From the limitations given in the discussion, it seems like this method is missing cases where the title is changed from the preprint to the accepted journal version, e.g. due to reviewer requests. Please add a sentence here whether it is possible, and if yes, how to mitigate for such cases.

Table 3: The entry for BioHackrXiv is inconsistent with the other (osf) preprints. Presumably the entry for the Versioning type should be “Single identifier for all versions”. Please, double-check and correct if necessary.

Lines 257-258: Unless there are legal reasons to completely remove entries from the database, it would be preferable to retain the original data, for the benefit of meta-research, e.g. into research misconduct. Are the reasons for removal taken into consideration? Perhaps one or two clarifying sentences would be useful here.

Reviewer #2: I enjoyed reading this manuscript that presents the Europe PMC open database of life science literature. It provides a comprehensive account of the motives and processes implemented and, when published, should become a go-to reference for anyone interested in ‘preprint metasearch’ --- I'm suggesting the authors to consider referring to this concept (https://en.wikipedia.org/wiki/Metasearch_engine) that was popular in the years 1995-2005.

Here are my comments:

* Rationale behind the naming

Line 123 reads:

“Europe PMC (https://europepmc.org/) is an open science platform and a life science literature database [35]. It is supported by a group of 35 international science funders as their repository of choice. Europe PMC is developed by the European Bioinformatics Institute (EMBL-EBI). It is part of the PubMed Central (PMC) International archive network, built in collaboration with the PMC archive in the USA. Europe”

Could the authors explain why Europe appears in the name? I understand there's a connexion to the developer (EMBL-EBI) ... but is there a particular focus on the indexing of European research?

* Preprint-Publication Linking (line 125)

Preprint-article matching is not a trivial process. The authors sketch up how their process work. Could they be a bit more specific and compare to other research addressing this information linking task (Cabanac et al., 2021; Eckmann & Bandrowki, 2023)? These 2 papers report how they perform on benchmarks, how does Europe PMC compare?

* Post-Publication Peer Review: PubPeer

Table 4 lists preprint review platforms. I believe PubPeer is missing (PubPeer hosts comments on preprints, too). See for instance https://pubpeer.com/search?q=SSRN

* Success stories

The authors thought about commenting some ‘success stories’ that Europe PMC has supported (lines 631 to 652). This is useful and should be stressed with a few more examples. Which community does Europe PMC serves, why no other option could replace the Europe PMC corpus? Why is this infrastructure critically important in some research areas?

* Non Crossref DOIs

Line 666 reads: ‘some do not’. Could the author try and quantify ‘some’. I suppose that's a minority ... and this should be clarified to understand what's missing in terms of coverage here. I saw the list of repositories on line 733 but failed to grasp the ‘magnitude of the loss’ (percentage of all preprints, say).

* Suggestions

1. In Table 1, add an estimate of the number of preprints hosted at each server

2. line 257 reads ‘and not even a notification remains.’ Indeed, this has been documented for SSRN for instance: https://retractionwatch.com/2021/05/17/preprints-are-works-in-progress-the-tale-of-a-disappearing-covid-19-paper/

3. Biological annotations (l290): How about RRIDs? https://scicrunch.org/resources

4. line 335: Which search engine technology runs Europe PMC? (e.g., Lucene, Solr, Elasticsearch?)

5. line 395: on publications following a preprint, one should perhaps acknowledge that some preprints will stay as such and no journal article will follow (because the authors decide not to).

6. line 1028: extra space to be removed: ‘Covid- 19’

7. Figure 5: add the percentage above each bar.

* References

Cabanac, G., Oikonomidi, T., & Boutron, I. (2021). Day-to-day discovery of preprint–publication links. In Scientometrics (Vol. 126, Issue 6, pp. 5285–5304). https://doi.org/10.1007/s11192-021-03900-7

Eckmann, P., & Bandrowski, A. (2023). PreprintMatch: A tool for preprint to publication detection shows global inequities in scientific publication. In PLOS ONE (Vol. 18, Issue 3, p. e0281659). https://doi.org/10.1371/journal.pone.0281659

6. PLOS authors have the option to publish the peer review history of their article (what does this mean?). If published, this will include your full peer review and any attached files.

Reviewer #1: **Yes: **Vladislav Nachev

Reviewer #2: No

---

## [Author Response · Author response to Decision Letter 0]

6 Aug 2024

The authors of this manuscript sincerely thank reviewers and the editor for their valuable time and effort. Your constructive notes have helped us improve this publication. We appreciate review of the code and data along with the manuscript itself. 

We have carefully considered each comment and have made every effort to address them thoroughly. Below, we provide point-by-point responses to the reviewers' comments.

We would also like to bring to your attention the change to the manuscript text regarding license restrictions for XML version of preprints posted to bioRxiv and medRxiv. The updated text within the ‘Technical challenges of indexing preprints’ section clarifies that the reuse of the XML version is governed by the license selected by the preprint authors. This change is brought on in response to preprint feedback that we have received and is also reflected in the second version of the preprint.

Response to Reviewer 1

- Lines 71-76: It was unclear from the description and analysis code whether the total number of articles in PubMed includes preprints or not. If it does, then it is appropriate to refer to the share and percentage of preprints. If not, then the measure is preprints per article and not a percentage, nor a share. Please clarify if the number of preprints is included in the total. Incidentally, when I ran the code, the result was exactly 12 and not over 12, but this is probably due to updates in the database.

Response: Thank you for pointing this out. Indeed, in the submitted manuscript the number of articles did not include preprints and the graph was displaying the number of preprints per article. We have changed the underlying code to calculate preprint percentage by dividing the number of preprints in Europe PMC posted in a given year by the sum of preprints and PubMed journal articles published in the same period. This change is reflected in the figure, figure legend, and in the manuscript text (line 72). 

Regarding your note on percentage changes when running the code - this is indeed due to database updates. 

- Figure 2 and lines 90-91: The numbers were higher than 30 only for the first two months, so it would be more precise to replace “few” with “two”. Using the code provided on zenodo, it seems that between 2 and 6% of the journal articles were associated with a preprint (hasPreprint == TRUE). It would be nice to reflect this in this figure, by splitting the articles into articles associated with a preprint or not. This way the conversion of preprints to journal articles becomes more apparent and complements the analysis in Fig. 6.

Response: We have changed “few” to “two” in the sentence on line 90.

Thank you for your suggestion to depict the number of journal articles based on a preprint in Figure 2. We have added a new panel to the figure that indicates the share of COVID-19 journal articles that were associated with a preprint.

- Line 112: From a security and privacy standpoint it is best to avoid using link shorteners, as they obscure the domain of the landing URL. Please replace with the original URL instead.

Response: Replaced with original URL as suggested.

- Lines 206-210: For better methods reproducibility it would be good to know which tools and external vendors are used for the conversion of PDF to JATS XML and to HTML.

Response: Thank you for your suggestion. We named the external vendor used for XML conversion and added a link to the open source XSL stylesheet used in-house to create the HTML version of the preprint.

- Lines 230-231: From the limitations given in the discussion, it seems like this method is missing cases where the title is changed from the preprint to the accepted journal version, e.g. due to reviewer requests. Please add a sentence here whether it is possible, and if yes, how to mitigate for such cases.

Response: Indeed existing preprint-article links can be missed by our method due to title changes. We have edited the text within the ‘‘Results’ section to clarify the limitation of the current matching algorithm, specifically related to title and first author changes (lines 413-415). We have also added a paragraph to the ‘Enriching preprint metadata’ subsection of the Discussion section to describe our plans to utilise a forthcoming automated matching mechanism developed by Crossref, which would improve matching coverage (lines 783-792).

- Table 3: The entry for BioHackrXiv is inconsistent with the other (osf) preprints. Presumably the entry for the Versioning type should be “Single identifier for all versions”. Please, double-check and correct if necessary.

Response: Thank you for pointing this out. We have corrected the entry for BioHackrXiv in Table 3 to “Single identifier for all versions”.

- Lines 257-258: Unless there are legal reasons to completely remove entries from the database, it would be preferable to retain the original data, for the benefit of meta-research, e.g. into research misconduct. Are the reasons for removal taken into consideration? Perhaps one or two clarifying sentences would be useful here.

Response: Thank you for your question. Europe PMC only removes records for preprints that have been completely erased from the corresponding server. We are not aware of the reason for deletion unless the author discloses this when contacting the Europe PMC team. We have added a clarification to the ‘Tracking changes to preprints’ subsection of the Results section (lines 450-455).

Response to Reviewer 2

- I'm suggesting the authors consider referring to ‘preprint metasearch’ (https://en.wikipedia.org/wiki/Metasearch_engine), a concept that was popular in the years 1995-2005.

Response: Thank you for your suggestion. We have referred to Europe PMC as a metasearch solution for preprints on the line 323.

- Rationale behind the naming. Line 123 reads: “Europe PMC (https://europepmc.org/) is an open science platform and a life science literature database [35]. It is supported by a group of 35 international science funders as their repository of choice. Europe PMC is developed by the European Bioinformatics Institute (EMBL-EBI). It is part of the PubMed Central (PMC) International archive network, built in collaboration with the PMC archive in the USA. Europe”

Could the authors explain why Europe appears in the name? I understand there's a connection to the developer (EMBL-EBI) ... but is there a particular focus on the indexing of European research?

Response: Thank you for your question. Europe PMC is a global repository and the content it hosts is not restricted to European research.

The rationale behind the naming is related to the deposition services that Europe PMC provides to authors supported by its funders. Europe PMC was originally launched as UKPMC - a dedicated archive for full text open access publications supported by UK-based biomedical research funders. In 2012 the database name was changed to Europe PMC due to expansion of its funding consortium and inclusion of funding organisations based in Europe, such as the ERC, FWF, and Telethon Italy. Currently Europe PMC is supported by a group of 35 international science funders, including some that are not based in Europe or UK (such as WHO, India Alliance, and EDCTP).

To clarify this we have made changes to the text in the ‘Introduction’ section (lines 129-136).

- Preprint-Publication Linking (line 125). Preprint-article matching is not a trivial process. The authors sketch up how their process work. Could they be a bit more specific and compare to other research addressing this information linking task (Cabanac et al., 2021; Eckmann & Bandrowki, 2023)? These 2 papers report how they perform on benchmarks, how does Europe PMC compare?

Response: We have added further details of the logic behind the matching process to the ‘Linking preprints to published journal articles’ subsection of the ‘Methods’ section (lines 241-246). We have described our plans to utilise a forthcoming Crossref matching mechanism, as well as plans to expand the links to include journal articles not indexed in PubMed in the ‘Enriching preprint metadata’ subsection of the Discussion section (lines 783-792).

Originally, our preprint-publication matching algorithm was developed back in 2018. At that time no similar algorithms were openly available for life science preprints to our knowledge to benchmark against. 

From the beginning we have set a very high threshold for preprint-publication matching accuracy. As an archive of scholarly information it is of utmost importance that the information we share is trustworthy. Crossref metadata supplied by preprint servers and journal publishers is the primary source for preprint-publication links, and our matching mechanism has been designed to supplement that. Therefore we do not feel that benchmarking our existing algorithm against those published more recently will be beneficial, as it has been intentionally designed to be very conservative with impact on sensitivity.

- Post-Publication Peer Review: PubPeer

Table 4 lists preprint review platforms. I believe PubPeer is missing (PubPeer hosts comments on preprints, too). See for instance https://pubpeer.com/search?q=SSRN

Response: Thank you for pointing this out. It is correct that PubPeer is not included in Table 4. Preprint review information is retrieved from Crossref, Sciety and Early Evidence Base. Metadata for PubPeer reviews is not shared via these services and therefore is not available in Europe PMC. As Table 4 only lists preprint review groups with review information available in Europe PMC, we have not added PubPeer to the list.

- Success stories. The authors thought about commenting some ‘success stories’ that Europe PMC has supported (lines 631 to 652). This is useful and should be stressed with a few more examples. Which community does Europe PMC serves, why no other option could replace the Europe PMC corpus? Why is this infrastructure critically important in some research areas?

Response: Thank you for your suggestion as well as your questions. Throughout the publication we highlight different ways in which data and services provided by Europe PMC are used. Importance of Europe PMC as an open infrastructure for the preprint community is explained in the Introduction, within the ‘Facilitating preprint discovery with Europe PMC’ section. Europe PMC offers a free and open metasearch solution for life science preprints across many different servers for both website and programmatic users. It provides sustainable and transparent community-governed infrastructure, focussed on addressing user needs. 

As an open life science resource Europe PMC supports many different user communities, from academic researchers, to clinicians, librarians, developers, text-miners, curators and members of the general public (Davé A, Seth V, Pottinger E, Rosenberg C, Potau X, Varnai P. Measuring the Value and Impact of the Europe PMC Repository [Internet]. Wellcome Trust; 2019. Available from: https://wellcome.figshare.com/articles/online_resource/Measuring_the_Value_and_Impact_of_the_Europe_PMC_Repository/8326559/2).

We highlight Europe PMC’s importance for the systematic review community in ‘Supporting preprints inclusion in meta-analyses and systematic reviews’ section within Discussion and reference multiple studies that have used Europe PMC preprint search for evidence synthesis. We describe the impact of Europe PMC services for text-mining and programmatic analyses in the ‘Providing programmatic access for preprints’ section and reference several use cases.

To incorporate your suggestion, we have expanded the latter section to include another use case of Europe PMC API from Sciety, a platform for reviewing and curating preprints in the life sciences (lines 665-666). We have also described our outreach efforts to increase the number of preprints with license information in Crossref, supporting programmatic preprint reuse and their inclusion by other discovery services (lines 677-687).

As an open scholarly infrastructure Europe PMC does not impose any barriers to access its APIs and bulk downloads. Users do not need to register or sign in to retrieve data through these services. This makes it difficult to track Europe PMC use. Use cases cited in this publication were brought to our attention either through personal communication or through literature search for publications citing use of Europe PMC.

While we are aware of other use cases, including Zenodo, Dimensions, or OpenCitations (https://europepmc.org/API-case-studies), they are not specific to preprints. Therefore we chose to omit them when writing this article.

- Non Crossref DOIs. Line 666 reads: ‘some do not’. Could the author try and quantify ‘some’. I suppose that's a minority ... and this should be clarified to understand what's missing in terms of coverage here. I saw the list of repositories on line 733 but failed to grasp the ‘magnitude of the loss’ (percentage of all preprints, say).

Response: Thank you for your question. We are not aware of any continuously updated community resource that tracks life science preprints servers. Without a complete list of relevant servers we cannot quantify how many of them do not register Crossref DOIs. 

To ensure better coverage, we continually check for new servers that should be added to Europe PMC using community resources, such as preprint archives content coverage in search services (Scholarly search engine comparison) and list of preprint servers from ASAPbio (https://asapbio.org/preprint-servers). If a server does not register Crossref DOIs, but fulfils other inclusion criteria, including free API access, we would consider including it in Europe PMC.

To clarify this in our publication we have made text changes to the corresponding paragraph and added examples of preprint servers that contain some life science preprints and do not register Crossref DOIs (lines 699-704).

- In Table 1, add an estimate of the number of preprints hosted at each server

Response: Thank you for your suggestion. The number of preprints for each server included in Europe PMC as of 15 September 2023 is available in our previous publication (Rosonovski S, Levchenko M, Bhatnagar R, et al. Europe PMC in 2023. Nucleic Acids Research. 2024 Jan;52(D1):D1668-D1676). As the number of preprints in the database changes daily, we provide the search syntax for preprints from each server in Table 1. Readers can use the search syntax to get an accurate number of preprints at any point in the future.

- line 257 reads ‘and not even a notification remains.’ Indeed, this has been documented for SSRN for instance: https://retractionwatch.com/2021/05/17/preprints-are-works-in-progress-the-tale-of-a-disappearing-covid-19-paper/

Response: Thank you for bringing this to our attention.

- Biological annotations (l290): How about RRIDs? https://scicrunch.org/resources

Response: RRID mentions in preprints are routinely identified using text-mining and are available as a sub-type of accession number annotations. See results for (ACCESSION_TYPE:rrid) AND (SRC:PPR) search query that brings up preprints mentioning RRIDs.

- line 335: Which search engine technology runs Europe PMC? (e.g., Lucene, Solr, Elasticsearch?)

Response: We have added a sentence to indicate that Europe PMC uses the Solr platform as the underlying search engine technology (lines 350-351).

- line 395: on publications following a preprint, one should perhaps acknowledge that some preprints will stay as such and no journal article will follow (because the authors decide not to).

Response: We have added a sentence to the beginning of the paragraph to explain that some preprints are intended as the final version of record (lines 401-402).

- line 1028: extra space to be removed: ‘Covid- 19’

Response: Thank you for pointing this out. At present there is no extra space in the term ‘Covid-19’ on the line 1028.

- Figure 5: add the percentage above each bar.

Response: Thank you for your suggestion. We have added a percentage for preprints with more than one version to the text (line 394). We did not add percentages to the bars in Figure 5, as the graph only displays the number of preprint records that have more than one version in Europe PM

---

## [Decision Letter · Decision Letter 1]

9 Sep 2024

Enabling preprint discovery, evaluation, and analysis with Europe PMC

PONE-D-24-15144R1

Dear Dr. Levchenko,

We’re pleased to inform you that your manuscript has been judged scientifically suitable for publication and will be formally accepted for publication once it meets all outstanding technical requirements.

**I would like to thank the 2 reviewers **for their help in assessing the manuscript and to thank you for addressing carefully all their comments. As you will see, the reviewers and I were satisfied with your edits. The manuscript is in a good shape now.

Kind regards,

Florian Naudet, M.D., M.P.H., Ph.D.

Academic Editor

PLOS ONE

Additional Editor Comments (optional):

Reviewers' comments:

Reviewer's Responses to Questions

**Comments to the Author**

1. If the authors have adequately addressed your comments raised in a previous round of review and you feel that this manuscript is now acceptable for publication, you may indicate that here to bypass the “Comments to the Author” section, enter your conflict of interest statement in the “Confidential to Editor” section, and submit your "Accept" recommendation.

Reviewer #1: All comments have been addressed

Reviewer #2: All comments have been addressed

2. Is the manuscript technically sound, and do the data support the conclusions?

Reviewer #1: Yes

Reviewer #2: Yes

3. Has the statistical analysis been performed appropriately and rigorously? 

Reviewer #1: N/A

Reviewer #2: Yes

4. Have the authors made all data underlying the findings in their manuscript fully available?

Reviewer #1: Yes

Reviewer #2: Yes

5. Is the manuscript presented in an intelligible fashion and written in standard English?

Reviewer #1: Yes

Reviewer #2: Yes

6. Review Comments to the Author

Reviewer #1: To my reading all comments from all previous reviewers have been appropriately addressed.

The article describes a valuable resource for the (meta-)research community.

Reviewer #2: Thanks for addressing my comments and for providing a point-by-point response.

7. PLOS authors have the option to publish the peer review history of their article (what does this mean?). If published, this will include your full peer review and any attached files.

Reviewer #1: **Yes: **Vladislav Nachev

Reviewer #2: No

---

## [Editor Report · Acceptance letter]

16 Sep 2024

PONE-D-24-15144R1 

PLOS ONE

Dear Dr. Levchenko, 

I'm pleased to inform you that your manuscript has been deemed suitable for publication in PLOS ONE. Congratulations! Your manuscript is now being handed over to our production team.

Kind regards, 

on behalf of

Pr. Florian Naudet 

Academic Editor

PLOS ONE